# One-dimensional extended Hubbard model with soft-shoulder potential

Thomas Botzung[1*], Guido Pupillo[1], Pascal Simon[2], Roberta Citro[3, 4] and Elisa Ercolessi[5, 6]

**1** IPCMS (UMR 7504) and ISIS (UMR 7006), University of Strasbourg and CNRS, 67000 Strasbourg, France
**2** Université Paris-Saclay, CNRS, Laboratoire de Physique des Solides, 91405, Orsay, France.
**3** Dipartimento di Fisica "E.R. Caianiello", Universit'a di Salerno and Spin-CNR,Via Giovanni Paolo II, 132, I-84084 Fisciano (Sa), Italy
**4** INFN, Sezione di Napoli, Gruppo collegato di Salerno, I-84084 Fisciano (Sa), Italy
**5** Dipartimento di Fisica e Astronomia dell'Università di Bologna, I-40127 Bologna, Italy
**6** INFN, Sezione di Bologna, I-40127 Bologna, Italy
*thomas.botzung@etu.unistra.fr

July 10, 2020

## Abstract

We investigate the $T = 0$ phase diagram of a variant of the one-dimensional extended Hubbard model where particles interact via a finite-range soft-shoulder potential. Using Density Matrix Renormalization Group (DMRG) simulations, we evidence the appearance of Cluster Luttinger Liquid (CLL) phases, similarly to what first predicted in a hard-core bosonic chain [M. Mattioli, M. Dalmonte, W. Lechner, and G. Pupillo, Phys. Rev. Lett. 111, 165302]. As the interaction strength parameters change, we find different types of clusters, that encode the order of the ground state in a semi-classical approximation and give rise to different types of CLLs. Interestingly, we find that the conventional Tomonaga Luttinger Liquid (TLL) is separated by a critical line with a central charge $c = 5/2$, along which the two (spin and charge) bosonic degrees of freedom (corresponding to $c = 1$ each) combine in a supersymmetric way with an emergent fermionic excitation ($c = 1/2$). We also demonstrate that there are no significant spin correlations.

# 1 Introduction

It is well known that quantum effects are very relevant in low dimensions. In particular, at low temperatures and in one dimension (1D), quantum fluctuations can become important enough to prevent the conventional Spontaneous Symmetry Breaking mechanism and influence the zero-temperature ($T = 0$) phase diagram of a model [1]. For fermions, the standard Fermi liquid picture fails and is substituted by the new paradigm of the Tomonaga-Luttinger Liquid (TLL) [2]. The standard difference between bosons and fermions is blurred, since it is known that there exist exact mappings of the free/weakly interacting fermionic theories into free/weakly interacting bosonic ones [2–7], with an underlying compactified boson Conformal Field Theory [8]. When interactions are introduced, the relevant model that appears is the so-called sine-Gordon model [9, 10], which is still an integrable model [4]. In 1D, there is another striking phenomenon that can be predicted theoretically and is by now observed in a variety of experimental set-up [11–16], namely the separation of charge and spin degrees of freedom [2, 17] that can propagate independently along a chain.

Such a picture is the key to understand the low temperature behaviour of the majority of both spin and fermionic quantum 1D model, at least as long as interactions are short-ranged such as in the Hubbard model and its extensions. It is therefore relevant for the majority of condensed matter systems in 1D, so diverse as organic materials [18], nanowires [19], carbon nanotubes [20], edge states in quantum Hall materials [21, 22], By now, this behaviour has also been realised in artificial materials with cold gases of atoms [23, 24], ions [25–27] and molecules [28, 29]. However, very recently possible signatures of a breakdown of the conventional TLL theory have been reported in several contexts, such as when non-linear effects are included [30, 31], symmetry protected phases may arise [32, 33] or the Hamiltonian might admit emergent low-energy fermionic modes [34, 35].

It is thus important to investigate to what extent the conventional TLL paradigm is robust towards various different types of interactions. Recently, some of us have considered [36–38] the zero-temperature phases of a hard-core bosonic gas confined to 1D and interacting via a class of finite-range soft-shoulder potentials, extending over a few sites. It was shown that, for sufficiently high interactions, the system exhibits a critical quantum liquid phase with

qualitatively new features with respect to TLL. The emergence of such a phase is due to a combination of commensurability and frustration effects that concur to stabilize the formation of a new liquid phase, in which the fundamental elements are not the original hard-core bosons but rather clusters of elementary particles. Similar phenomena of self-assembled of composite objects has been studied in a variety of other models, also in higher dimensions, such as colloidal particles and polymers [39–41] and ultracold atoms and molecules [42]. In some cases, the competition between superfluidity and clustering might lead to the formation of supersolids [42–48].

In this paper, we consider an extended version of the Hubbard model on a chain, in which a system of spinful fermions can interact via both an on-site $U$-potential and an off-site repulsion of strength $V$, which is finite within a spatial range of length $r_c$. The case $r_c = 1$ is known as the extended Hubbard model and it has been much studied in the case of half-filling ($\rho = 1/2$) for its interesting phase diagram. The latter displays a bond-order (BOW) phase for $U \sim V$, which separates a charge-density wave (CDW) phase ($U > V$) from a spin-density wave (SDW) phase ($V > U$). A complete study of this model and of the so called correlated-hopping extended Hubbard model has been presented in [49], where the use of bosonisation techniques complemented by a renormalization group analysis has allowed for a complete classification of possible phases. As mentioned above for generic 1D systems, in that work quantum phases are characterized only in terms of the dominant correlation functions and their discrete symmetries. Recently, this model has been re-examined in light of symmetry protected topological orders [50] and a complete classification of its phases in terms of nonlocal, parity and string-like, order parameters has been given. Similar results have been obtained also for the half-filled dipolar gas chain [51], i.e. with long-range interactions decaying as the cube of the distance. With the exception of the case of quarter-filling, where commensurate effects can still play a role, the general case of small lattice filling (i.e., below one half) has received little attention, since it is understood that the presence of empty sites makes the system trivially metallic, with no "exotic" phases. In these cases, the model is adequately described by means of the TLL theory.

The situation can be drastically different, however, if we consider a soft-shoulder potential with a finite range $r_c > 1$. As we will see, commensurability between particle density and the range of the potential may lead to the formation of clusters of particles that are free to move as a whole, leading to new types of gapless phases that we can define as a TLL of clusters (CLL phases). As the interaction strength parameters $U, V$ change (both potentials are assumed to be repulsive, $U, V > 0$), we encounter different types of clusters, originating different types of CLL phases. The possibility to observe some of these CLL effects in experiments with cold Rydberg atom systems has been considered in [37].

The phase diagram for temperature $T = 0$ of the extended Hubbard model with repulsive soft-shoulder interactions, for a potential range $r_c = 2$ and a fermionic density of $\rho = 2/5$, is presented in Fig. 3. It summarises the results of our numerical investigation, performed by means of a Density Matrix Renormalisation Group (DMRG) algorithm [52–55]. Due to the high degree of frustration which is present in the model and to the fact that the system is always in a critical phase, reaching a high precision in numerical simulations is a very challenging endeavour. Despite so, by combining different indicators -such as charge and spin structure functions, spin/charge/single-particle gaps, von-Neumann entanglement entropy- we are able to completely characterise the different phases. We demonstrate that the stan-

dard TLL phase is the ground state for low values of $U, V$ only. For large $U$, we encounter a CLL phase in which clusters contain either a single particle or a couple of nearest neighbour ones. For large $V$, instead, we have a doublon phase described by clusters with either a single particle or double occupied sites. We denote these two phases with $\mathrm{CLL}_{nn}$ and $\mathrm{CLL}_d$ respectively. At the semiclassical level, we find that the transition between these two liquids occurs at $V = 2U/3$. All these phases are characterized by a central charge $c = 2$. However, in the $\mathrm{CLL}_{nn}$ we show that the single-particle excitation is gapped; hence we interpret this phase as a TLL made of composite clusters particles. Interesting, the $\mathrm{CLL}_{nn}$ phase is separated from the TLL phase by a critical line with central charge $c = 5/2$. By examining the sound speed of the different excitations, we can conclude that at these points there is an emergent supersymmetry, with the two bosonic degrees of freedom (spin and charge, with $c = 1$ each) combining with a emergent fermionic one, with $c = 1/2$. Finally, we notice that, for very low $U$ and large/intermediate $V$, we find a tendency of the system to form a liquid of only double occupied sites.

The paper is organized as follows. We start by introducing the model in Sec. 2. We then focus on the classical limit $(t = 0)$ of our model in Sec. 2.1, which provides physical insights on the frustration mechanisms at play and allows to predict some of the phases in our system. In Sec. 2.2, we introduce the observables, such as the static structure factor, the low-energy degrees of freedom and the entanglement entropy, which are used to describe the ground state of the quantum model. In Sect. 3, we analyse in details the different phases in the whole-range of the parametera $U$ and $V$, by computing numerically the different observables introduced in the previous section, leading to the complete phase diagram of the model. In particular, we demonstrate that the ground state corresponds to a standard TLL phase for low values of $U$ and $V$, while it consists of new types of liquids made of clusters of particles for large $U$ and large $V$, in agreement with the classical prediction (Secs. 3.1 and 3.2). We investigate in details the transition between the different phases for an intermediate value of $U$ in Sec. 3.3, and in Sec. 3.4 we show that, for very low $U$ and intermediate $V$, the system tends to form a liquid phase that features more doubly occupied sites than the standard TLL phase and is not captured by the classical approximation. Details on the numerical analysis are provided in Appendix A and B.

## 2 The model

In this work, we consider an extended Hubbard chain with on-site and finite-range soft-shoulder repulsion, described by the Hamiltonian

$$\mathcal{H} = -t \sum_{i,\sigma} (c_{i\sigma}^\dagger c_{i+1\sigma} + \mathrm{h.c}) + U \sum_i n_{i\uparrow} n_{i\downarrow} + V \sum_i \sum_{\ell=1}^{r_c} n_i n_{i+\ell}, \tag{1}$$

where $c_{i\sigma}^\dagger$, $c_{i\sigma}$ are creation/annihilation operators of fermionic particles with spin $\sigma = \uparrow, \downarrow$ on the site $i$ and $n_i = n_{i,\uparrow} + n_{i,\downarrow}$. The coefficient $t$ represents the tunneling matrix element (and will be taken to be unitary in the following $t \equiv 1$), while $U$ gives the strength of the on-site interaction between two fermions on the same site (and opposite spin) and $V$ that of the soft-shoulder density-density interaction. This density-density interaction yields a contribution $V$ only if two occupied sites lie within a distance $r_c$ of each other, and zero otherwise. Here, we

assume only purely repulsion interactions with $U, V > 0$ and impose antiperiodic boundary condition. The physics of this model is determined also by an additional implicit parameter, the particle density $\rho = N/L$, $N$ and $L$ being the total number of particles and the size of the chain respectively.

## 2.1 Classical analysis

Before going into the detailed analysis of the quantum phase diagram of Eq. (1), it is convenient to first consider the exact solution of the ground state in the classical limit $(t = 0)$. In this regime, the main physics can be inferred by looking at the competition between the soft-shoulder potential, the on-site interaction and the two relevant length scales, namely the cut-off radius $r_c$ and the mean spacing between particles $r^* = 1/\rho$, where $\rho = N/L$ is the particle density ($N$ and $L$ are the total number of particles and the size of the chain, respectively).

Let us first consider the regime $U \gg V$ which prevents double occupancy. In this case, the model becomes substantially the same as a spinless fermionic model. It was shown in Ref. [36] that the latter exhibits a rich phase diagram with three different phases:
(i) For $r^* > r_c$, the particle density is small and the ground state corresponds to a liquid type phase.
(ii) For $r^* = r_c$, particles are equally spaced every $r_c + 1$ lattice sites, corresponding to a crystalline order.
(iii) The most interesting situation occurs for $r^* < r_c$, where the competition between $r^*$ and $r_c$ leads to frustration and particles have tendency to self-assemble into clusters (blocks). These effects result in the formation of a highly degenerate ground state because these different types of clusters can be assembled in many different ways, thus leading to a liquid of clusters.

In order to describe the classical ground state of (iii) in a one-dimensional system, we extend the *cluster exchange model* introduced in [36]. The key idea is to identify the cluster of particles and holes in the configuration with lowest energy. First, we note that the block with zero energy consists of one particle followed by a number of $r_c$ of empty sites. We denote this cluster as A [see Fig. 1]. Secondly, we want to identify the blocks with the smallest finite energy. To do so, we have to take the competition between the on-site ($U$) and soft-shoulder ($V$) potentials into account. When $U \ll V$, the cluster made of a doubly occupied site followed by $r_c$ empty sites is favoured with an energy $U$. We refer to this cluster as B' [see Fig. 1]. When $U \gg V$, a cluster made by two nearest-neighbors followed by $r_c$ empty sites is energetically favoured and it costs an energy $V$. The latter is denoted as cluster B [see Fig. 1]. Other blocks with higher energy are discarded. Since exchanging two of these blocks let the energy unchanged, the classical ground state thus consist of all permutations of blocks A and B or A and B' (e.g. [ABAB],[AABB], or [AB'AB'],[AB'B'A]...). Introducing the total number of blocks $M$ for a system of length $L$ as

$$M = L(1 - \rho)/r_c, \tag{2}$$

the ground state degeneracy grows exponentially with $M$ (see below). Furthermore, the blocks themselves are degenerate due to the presence of the additional spin degree of freedom. This is responsible for an increase of the ground state degeneracy compared to the spinless case [see Fig. 1].

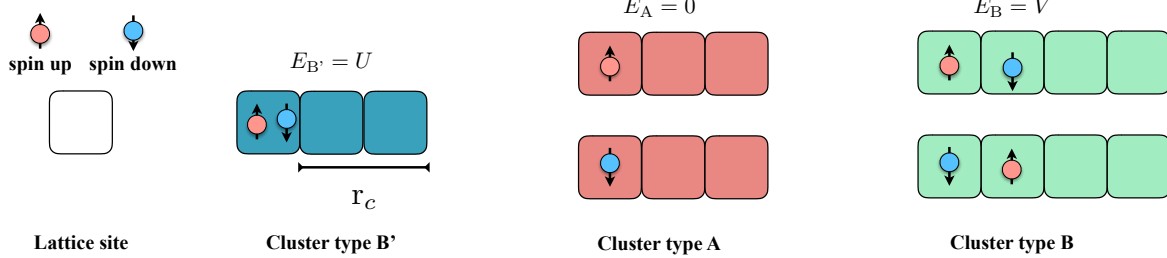

Figure 1: Graphical representation of the different possible blocks structure for the case $r_c = 2$.

In this work, we focus on the case $r_c = 2$ and choose a particle density $\langle n_\uparrow \rangle = \langle n_\downarrow \rangle = 1/5$. A convenient graphical representation illustrating the classical ground state configurations for $r_c = 2$ is presented in Fig. 2. This specific density corresponds to a ratio of $n_A$ block A and $n_B$ block B or $n_{B'}$ block B' equal to $n_B/n_A = n_{B'}/n_A = 1/2$. As already mentioned, the presence of the on-site interaction will slightly modify the classical picture. In particular, a transition can occur between a liquid made of clusters AB and a liquid made of clusters AB'. To gain physical insight, we define the energetical gain to transform all B $\to$ B' by $\Delta_{B \to B'} = n_B V - n_{B'} U$. When $\Delta_{B \to B'} < 0$ ($> 0$), B'(B)-clusters are favoured. One can note that the size of cluster B' ($r_c + 1$) is one site smaller than cluster B ($r_c + 2$). Thus, changing 3 cluster B to 3 cluster B' gives 3 vacant spaces, which corresponds precisely to the size of cluster A with zero energy. As a consequence, 3 B-clusters effectively transform into 2 B'-cluster and 1 A-cluster, lowering the energy by $\Delta_{B \to B'} = 3V - 2U$ and changing the ratio $n_{B'}/n_A = 1/4$. The transition is therefore found at $V = \frac{2}{3}U$. In the same manner one can find a generalization for arbitrary $r_c$ with the transition given by $V = \frac{r_c}{r_c+1}U$. From this argument, two situations can arise:

(i) $V < \frac{2}{3}U$, the cluster type B is favored with respect to the B'. The classical blocks configuration can be ordered in many different ways leading to an exponential degeneracy of the ground state $d = M!/[(M/3)!(2M/3)!]$, without considering the spin degeneracy. Fig. 2, panel (a), presents the corresponding cluster configuration.

(ii) $V > \frac{2}{3}U$, B'-clusters of double occupied sites are now favoured, with the same exponentially degenerate ground state. This case is presented in Fig. 2 panel(b).

For the rest of this chapter, we will denote with $CLL_{nn}$ the phase (i) with nearest-neighbor (B) clusters whereas the phase (ii) with doubly occupied sites (B' clusters) will be referred to as $CLL_d$. Clearly commensurability between the size of the clusters and the total size of the system is important. In the following analytical and numerical calculations we always assume that $L$ contains an integer number of clusters; therefore we will use $L = 20, 30, 40, 50, 60$.

## 2.2 Observables

Before investigating the influence of quantum fluctuations ($t = 1$) on the phase diagram in Sec. 3, we first introduce the different observables targeting both the spin and the charge sectors such as the structure factor, the low-energy degrees of freedom, and the entanglement entropy. In particular, we provide a classical interpretation of these observables when possible.

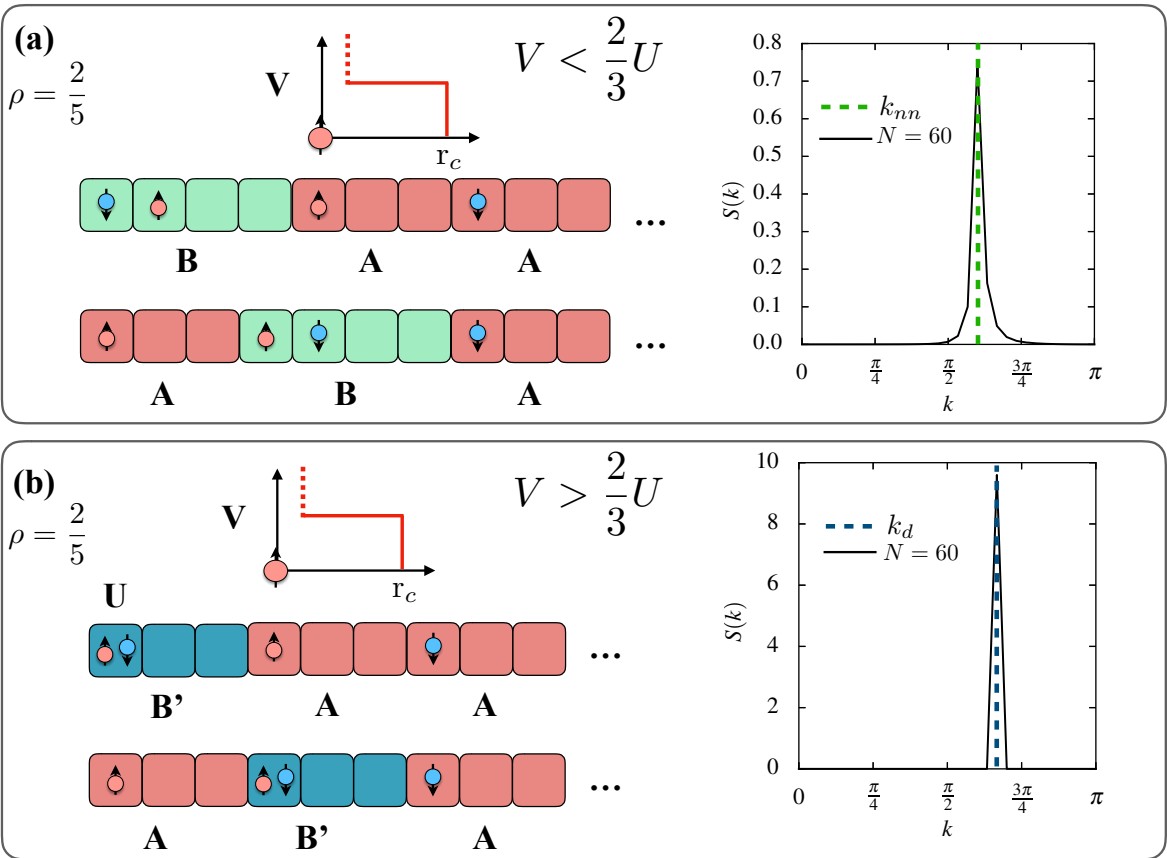

Figure 2: Graphical representation of a classical ground state configuration for the $\text{CLL}_{nn}$ phase in panel (a) and the $\text{CLL}_d$ phase in panel (b). As an example, we show in each panel, the corresponding peak in the structure factor.

**Structure factor**

Two-points correlations between particles or spins are known to give valuable information about the order and dynamics in condensed matter systems. The Fourier transform of the correlations in real space, namely the structure factor, reveals the relevant length scales, e.g., the periodicity along one and/or several axis arising from spontaneous symmetry breaking. In particular, a peak in the structure factor at a specific momentum indicates a precise spatial period. The charge and the spin structure functions are given by:

$$S_\nu(k) = \frac{1}{L} \sum_{\ell,j} e^{ik(l-j)} g_{2,\nu}(\ell - j)$$

with $g_{2,\nu}(\ell - j)$ the connected correlation function in the charge or spin sector ($\nu = c, s$), which reads:

$$
\begin{aligned}
g_{2,c}(\ell - j) &= \langle n_\ell n_j \rangle - \langle n_\ell \rangle \langle n_j \rangle \\
g_{2,s}(\ell - j) &= \langle S_\ell^z S_j^z \rangle - \langle S_\ell^z \rangle \langle S_j^z \rangle
\end{aligned}
\tag{3}
$$

We expect that the formation of the clusters phases should be well captured by the charge

structure factor. Indeed, the charge modulation in the classical phase already provides an estimate of the momentum peak's position of the charge structure factor. Within our classical approximation, we see in Fig. 2 that $S_c(k)$ exhibits a peak located at $k_{nn} = 2\pi M/L = 3\pi/5$ for the $\text{CLL}_{nn}$ [Fig. 2 (a)] and a peak at $k_d = 2\pi/3$ for the $\text{CLL}_d$ [Fig. 2 (b)] .

**Low-energy degrees of freedom**

The behaviour of the low-energy excitations is well captured by the charge and spin gaps defined as:

$$\Delta_c = E_{N+2}^{(\uparrow=\downarrow)}(L) + E_{N-2}^{(\uparrow=\downarrow)}(L) - 2E_N^{(\uparrow=\downarrow)}(L)$$

$$(4)$$

$$\Delta_s = E_N^{(\uparrow=\downarrow+2)}(L) - E_N^{(\uparrow=\downarrow)}(L)$$

where $E_N^{(\uparrow=\downarrow)}(L)$ is the ground state energy in the case of $N_\uparrow + N_\downarrow = L\rho$, $E_{N\pm2}^{(\uparrow=\downarrow)}(L)$ is the energy of the state obtained by adding/removing two particles with opposite spin and $E_N^{(\uparrow=\downarrow\pm2)}(L)$ is the energy of the state obtained by flipping the spin of one particle. Also, we consider the single particle gap:

$$\Delta_{sp} = E_{N+1}^{(\uparrow=\downarrow\pm1)}(L) + E_{N-1}^{(\uparrow=\downarrow\pm1)}(L) - 2E_N^{(\uparrow=\downarrow)}(L) \qquad (5)$$

where $E_N^{(\uparrow=\downarrow\pm1)}(L)$ is the energy of the state obtained by adding/removing one particle.

In the usual LL, all these gaps are expected to vanish. However, this is different in the cluster phase. In the following, we compute these gaps in the classical limit ($t = 0$) from the solution introduced in Sec. 2.1.

(i) In the $\text{CLL}_{nn}$ phase, we consider a system size of $L = 10\ell$, commensurate with the cluster formation, and a number of particles $n_A = 2\ell$ and $n_B = \ell$, where $\ell$ labels the number of building blocks. The classical energy of the system reads

$$E_N^{(\uparrow=\downarrow)}(L) = n_B V = \ell V. \qquad (6)$$

*(a) Single particle gap.* Now, upon adding/removing one particle we obtain

$$E_{N+1}^{(\uparrow=\downarrow\pm1)}(L) = \ell V + 2V \qquad E_{N-1}^{(\uparrow=\downarrow\pm1)}(L) = (\ell-1)V \qquad (7)$$

as the states cannot rearrange properly due to the frustration. Thus, the single particle gap is given by

$$\Delta_{sp} = \ell V + 2V + (\ell-1)V - 2\ell V = V, \qquad (8)$$

which implies that the single particle gap is always open within the $\text{CLL}_{nn}$ phase.

*(b) Spin gap.* We see that upon a spin flip, classical energy remains unchanged. Consequently the spin gap is always closed in this phase.

*(c) Charge (cluster) gap.* Here, the situation is remarkably different, since extracting/adding two particles allows the system to rearrange properly. For instance, if we consider a ground state of the form BAABAABAABAA, extracting two particles changes 3 clusters B to 4 clusters A creating the new configuration AAAAAAAAABAA, lowering the energy

by an amount $3V$. Instead, the opposite process, i.e. doping with two particles, changes 4 clusters A to 3 clusters B, thus increasing the energy by $3V$:

$$E_{N+2}^{(\uparrow=\downarrow)}(L) = \ell V + 3V \qquad E_{N-2}^{(\uparrow=\downarrow)}(L) = V - 3V. \tag{9}$$

Therefore the contributions associated to the insertion and extraction of a single cluster gap cancel out.

(ii) In the $\text{CLL}_d$ phase, we consider a system size $L = 3 \times 10\ell$, commensurate with the cluster formation, and again we set $n_A = 2\ell$ and $n_B = \ell$, where $\ell$ labels the number of building blocks. This yields a classical energy:

$$E_N^{(\uparrow=\downarrow)}(L) = n_{\text{B}'}U = 2\ell U. \tag{10}$$

*(a) Single particle gap.* Unlike the $\text{CLL}_{nn}$ phase, in this case, adding/removing a particle cost exactly the same energy to the system, $U$. Since we choose $\langle n_\uparrow \rangle = \langle n_\downarrow \rangle$, this effect is independent of the spin considered, and thus the single particle gap vanishes.

*(b) Spin gap.* Due to the double occupancy, at first glance we expect spin effect to be relevant. Nevertheless, since the number of cluster A is twice the number of clusters B', the system can be (at least classically) rearranged to have both spin flip contribution exactly cancelling out. The spin gap is then also zero in this phase.

*(c) Charge (cluster) gap.* In the same manner as the single particle gap, both contribution will exactly cancel, leading to a vanishing cluster gap.

This investigation indicates that there must be a transition between the $\text{CLL}_{nn}$ phase and the TLL where the single-particle gap $\Delta_{sp}$ opens linearly with $V$, while the cluster gap and spin gap remain gapless. The second transition between the $\text{CLL}_{nn}$ and the $\text{CLL}_d$, predicted at $V = \frac{2}{3}U$ in Sec. 2.1, is characterized by the fact that single-particle gap closes while the other gaps remain unchanged. We will show in the next Sec. 3 that quantum fluctuations $(t \neq 0)$ lead to qualitatively similar physical properties.

| Gaps Phases | $\Delta_{sp}$ | $\Delta_c$ | $\Delta_s$ |
|---|---|---|---|
| TLL | 0 | 0 | 0 |
| $\text{CLL}_{nn}$ | $V$ | 0 | 0 |
| $\text{CLL}_d$ | 0 | 0 | 0 |

Table 1: Summary table of the expected values of the gap inferred from the classical analysis.

## Von Neumann entropy

Entanglement plays a fundamental role in the study of strongly correlated systems and is widely used to characterize their critical properties. In particular, a change in the ground state entanglement allows to understand and locate a quantum phase transition. The most common way to measure entanglement between two parts of a system is provided by the Von Neumann entropy $S_L(\ell)$, which is also called entanglement entropy. If we consider a system of

L sites that is divided into two subsystems A and B containing $\ell$ and $L - \ell$ sites, respectively, $S_L(\ell)$ is given by:

$$S_L(\ell) = -\mathrm{Tr}\rho_\ell \log \rho_\ell, \tag{11}$$

where $\rho_\ell$ is the reduced density matrix of the sub-interval $\ell$ with respect of the rest of the chain.

In the thermodynamic limit, we have generally two different behaviors:

1. For gapped phases $S_L(\ell)$ follows an area law, i.e. is proportional to the surface of the block $\ell$ which is thus constant in 1D.

2. In critical gapless phases, the entropy diverges logarithmically, $S_L(\ell) \sim \log(\ell)$. In particular, for a conformal invariant one-dimensional system with periodic boundary condition, the entropy satisfies the following universal scaling law [56]:

$$S_L(\ell) = \frac{c}{3} \ln \left[ \frac{L}{\pi} \sin(\pi\ell/L) \right] + a_0 + \mathcal{O}(1/\ell^\alpha), \tag{12}$$

where $L$ is the system size of the system, $c$ the central charge of the theory, $a_0$ a non-universal constant and $\ell$ the block length. There is also additional correction of the form $1/\ell^\alpha$. From Eq. 12, we can extract the value of the central charge of the underlying conformal field theory, which roughly speaking is a measure of the degrees of freedom of the system, e.g., for a boson $c = 1$ and for a fermion $c = 1/2$. Since the low-energy degrees of freedom of both the TLL and the CLL phases [36] are described by a conformal field theory, we use Eq. 12 to obtain the entanglement properties of the ground state. Note that the TLL theory predicts a separation of the spin and charge degrees of freedom, which implies a central charge $c = 2$.

## 3 The phase diagram

In the following, we study how the physical properties found in the previous section within the classical approximation are modified in the presence of the tunneling term $t \neq 0$. We use a Density Matrix Renormalization Group (DMRG) algorithm [52–55] to investigate the quantum phase diagram at zero temperature ($T = 0$) of the soft-shoulder Hubbard model (Eq. (1)) in the whole range of the parameters $U/t, V/t$ (where $t$ is set to unity), for a potential range $r_c = 2$ and a fermionic density of $\rho = 2/5$ (see Fig. 3). In Sec. 3.1, we first characterize the nearest-neighbor cluster phase ($\mathrm{CLL}_{nn}$) obtained in the large-$U$ limit. In Sec. 3.2, we then study the doublon cluster phase ($\mathrm{CLL}_d$) occuring for large $V$. In the next Sec. 3.3, we consider the intermediate value of $U = 10$ and study the different phase transitions (TLL $\rightarrow$ $\mathrm{CLL}_{nn}$ and $\mathrm{CLL}_{nn} \rightarrow \mathrm{CLL}_d$ ) occuring as $V$ is increased. Finally, Sec. 3.4 is devoted to the study of the $\mathrm{TLL}_d$ phase, found for small $U$ for all $V$. Due to the high degree of frustration inherent to the model, reaching a high precision in numerical simulations is very challenging. Details on the numerical analysis are provided in the appendices A and B.

### 3.1 The large $U$-limit and the nearest-neighbor cluster phase ($\mathrm{CLL_{nn}}$)

Classically, we have shown in Sec. 2 that the $\mathrm{CLL}_{nn}$ phase appears when the on-site interaction $U$ is on a factor $3/2$ larger than the soft-shoulder repulsion $V$. To observe this phase, we consider a regime with $U \gg V \gg t$, and we therefore expect the spin gap to be closed, as in the standard Hubbard model. Since, in this limit, double occupancy is strongly avoided, the charge sector of our models mimics essentially a spinless fermionic model with a density of

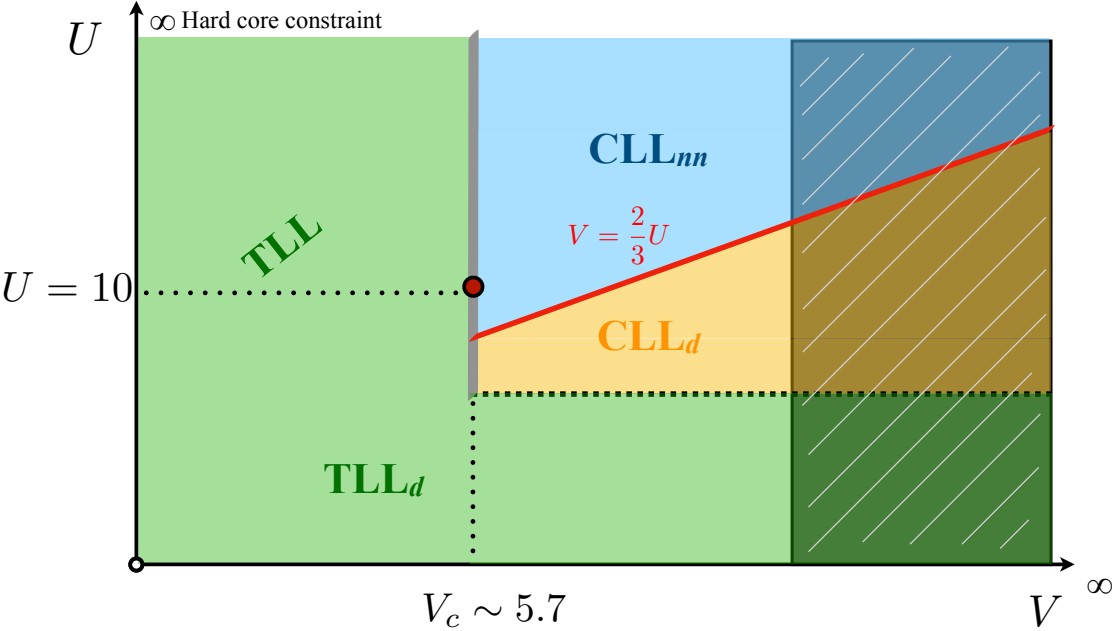

Figure 3: Sketch of phase diagram of the extended Hubbard model with soft-shoulder potential. We choose a potential range $r_c = 2$ and a fermionic density of $\rho = 2/5$. The green area represent the usual Tomonaga-Luttinger Liquid (TLL) phase and an anomalous Luttinger Liquid, which contains a large number of doubled occupied states (TLL$_d$). Here, CLL$_{nn}$ stands for Cluster Luttinger Liquid phase with nearest-neighbors cluster (blue) and CLL$_d$ for a CLuster Luttinger Liquid phase which contains doubled occupied states (yellow). The properties of these different phases are described in Sec. 2.1 and Sec.3. The red line shows to the classical result (see Sec. 2.1). The red dot for $V_c \approx 5.73$ corresponds to the critical point between TLL and CLL$_{nn}$ determined numerically for $U = 10$. The shaded area indicates where the numerical results are particularly hard to extract (around $V \approx 7$).

particles given by $\rho = \langle n_\uparrow \rangle + \langle n_\downarrow \rangle$, i.e., the model considered in Ref. [37]. In order to see that the formation of the block structure introduced in Sec. 2 survives with quantum fluctuation, we look at the numerical calculation of the (static) charge structure function $S_c(k)$ [1], which should develop a peak at $k_{nn} = 2\pi M/L$ where $M$ is the total number of clusters. In this case, denoted CLL$_{nn}$ in Sec. 2, we have $M/L = 3/10$, so that $k_{nn} = (3/5)\pi$. In Fig. (4) panel (a), we show the behaviour of $S_c(k)$ for $U = 50$ and different values of $V$. The emergence of the peak is evident.

## 3.2 The large $V$-limit and the doublon cluster phase (CLL$_d$)

For finite (but still large) $U$, it is necessary to take into consideration the fact that we have two species of fermions and that we might have double occupied sites. In the case $V \gg U \gg t$, the classical analysis [see Sec. 2.1] suggests that a second type of cluster phase might form.

---

[1]Clearly commensurability between the size of the clusters and the total size of the system is important. Here and in the following analytical and numerical calculations we assume that $L$ always contains an integer number of clusters.

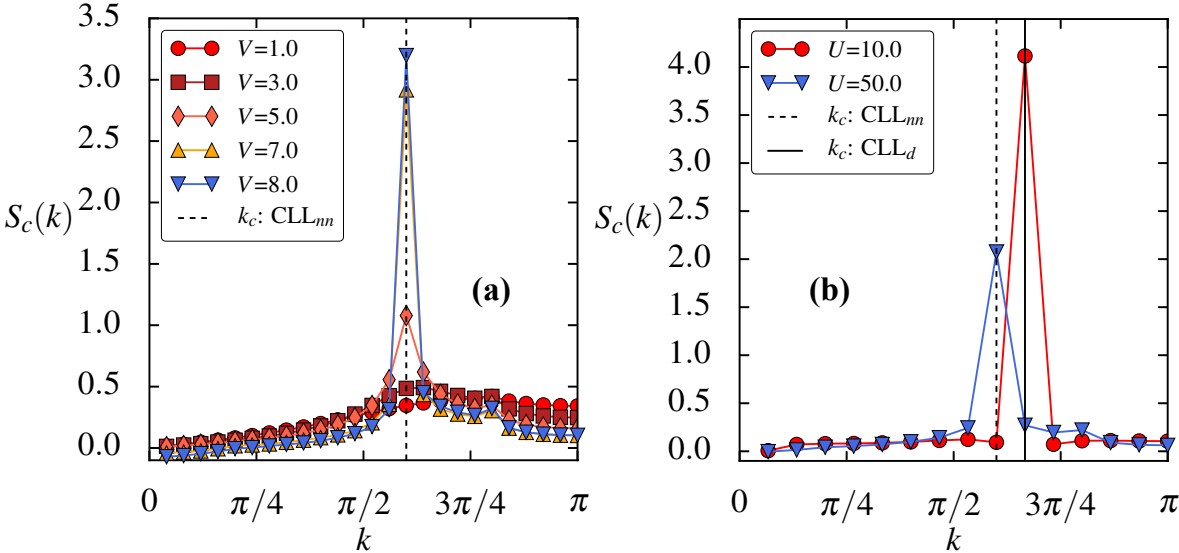

Figure 4: Panel (a): The charge structure function $S_c(k)$, evaluated for $U = 50$ and $V = 1, 3, 5, 8$. The numerical simulation is performed with $L = 50$, a size allowing for an exact number of clusters. Panel (b): The charge structure function $S_c(k)$, evaluated for $V = 8$ and $U = 10, 50$. The numerical simulation is performed with $L = 30$, a size allowing for an exact number of clusters for both the $\mathrm{CLL}_{nn}$ and the $\mathrm{CLL}_d$ phases.

Similarly to before, we have to respect a density constraint implying that $n_A = 4n_{B'}$ and we can order clusters of type $A$ and $B'$ in many different ways, leading to a liquid cluster phase which contains doubled occupied states. This phase was to the doublon cluster phase ($\mathrm{CLL}_d$) in Sec.2. A pictorial representation of it is shown in the panel (b) of Fig. 2. One can check that now $M/L = 1/3$ so that we expect to see a peak in the charge structure function at $k_d = 2\pi M/L = (2/3)\pi$.

At the semiclassical level ($t = 0$), the discontinuity in the energy between the $\mathrm{CLL}_{nn}$ and the $\mathrm{CLL}_d$ phases suggests a first order and appears at $3V = 2U$. Of course we expect strong renormalization effects due to the presence of the hopping term in the Hamiltonian, but the physical properties of the two phases should be the same. We can verify so, by examining what happens for an intermediate values of $V$, say $V = 8$ and two different values of $U$, one big (say $U = 50$) corresponding to a $\mathrm{CLL}_{nn}$ phase and one intermediate (say $U = 10$) corresponding to a $\mathrm{CLL}_d$ phase. The predicted shift of the peak in the charge structure function is indeed numerically verified as shown in Fig. (4), panel (b).

It is essential to notice that the position of the different peaks indicates a clear breaking-down of the TLL theory. Indeed, bosonization predicts a peak in the charge structure factor independently of $r_c$ at

$$k_c = 2\pi\rho \tag{13}$$

with $\rho$ the total density. For a hard-core particle model, it has been established [37] that this breakdown is rooted in the classical frustration inherent from this type of finite-range potential [see also Sec. 2.1]. Here, we have thus confirmed and extended this result to fermionic models, and we have demonstrated that the spin degree of freedom can lead to a novel type of cluster phase. To go further in the analysis, we would like to determine also if the presence of the

spin degree of freedom influences the transition between the TLL and the $\text{CLL}_{nn}$ phases.

## 3.3 The phases for intermediate values of $U$

In this subsection we fix the on-site interaction to the intermediate value $U = 10$. As mentioned previously, we expect now to find three phases: the TLL, the $\text{CLL}_{nn}$ and the $\text{CLL}_d$, in order of increasing values of $V$.

In the first place, we distinguish TLL, the $\text{CLL}_{nn}$ and the $\text{CLL}_d$ phases by looking at the peak in charge structure factor. In Fig. 5 (a), we find that the position $k_c$ of the maximum peak in $S_c(k)$ indicates the three phases. First, at small $V$, $k_c$ is located at $4\pi/5$ in agreement with a TLL liquid, see Eq. (13). For an intermediate $V$, $k_c = 3\pi/5$, corresponding to the classical charge modulation of the $\text{CLL}_{nn}$ (see Fig. 2). Finally at strong $V$, $k_c$ is equal to $\pi/3$, which corresponds to the $\text{CLL}_d$ phase (see Fig. 2). In general, a specific ordered phase is characterized by a finite non-zero value of $S_c(k_c)/L$ in the thermodynamical limit.

In order to see that this is the case, we have examined the finite size scaling of the charge structure factor $S_c(k)$. This is shown for instance in Fig. 5 panel (b) for several values of $V$. Dotted lines are best fits of the form $a + b/L + c/L^2$, where we have arbitrary considered size correction up to the second order. From these data, it is evident that $S_c(k)$ is finite in the thermodynamic limit for $V \geq 5.5$, whereas it goes to zero for smaller values of the interaction. In Fig. 5 panel (c), we present the extrapolated infinite size $S_c(k_c)$ value as a function of the interaction strength $V$.

To better understand the nature of the cluster phases, we now examine the spin structure function $S_s(k)$. By looking at Fig. 6 panel (a) and (b), we notice that, at small and intermediate $V$, the position $k_c$ of the maximum peak of $S_s(k)$ is located at $k_0 = (2/5)\pi$ in correspondence with the single-species density $\rho/2 = 1/5$ and the TLL theory (see Eq. (13)). For large values of $V$, instead, we found that the peak's position at $k_c = \pi$. We can clearly conclude that:

i) In the $\text{CLL}_{nn}$ phase AF order is enhanced. This result can be easily understood if we consider strong-coupling corrections to the large $U$ limit. Indeed, for infinite $U$ when double occupancy is strictly avoided and the hopping term can be neglected, the different spin sectors are exactly degenerate. For large but finite $U$, the spin configuration that allows for the maximum energy gain due to hopping is indeed the one in which two spins that are separated only by empty sites are AF ordered. However, this effect does not correspond to a true long range order, as it can be inferred from the fact that the peak at $k = \pi$ goes to zero (or a very small value) in the thermodynamic limit, e.g. see panels (b) and (c) of Fig. 6.

ii) In the $\text{CLL}_d$ phase the peak corresponding to the total density of (single) particles is shifted to smaller values of the momentum, signalling that there is a formation of a certain number of double occupied sites, effectively reducing the (single particle) spin density. To confirm the latter point, we study the doubly occupancy $d$, defined as:

$$d = \frac{1}{L}\sum_i n_{i,\uparrow} n_{i,\downarrow} - \rho, \tag{14}$$

where $n_{i,\sigma}$ is the occupation of one species of particles, and $\rho$ the total average density. Fig. 7 (a) shows the double occupancy as function of $V$. We find that the double occupancy increases continuously with $V$ up to a discontinuous jump around $V \sim 7.5$, which could indicate a first order transition between the $\text{CLL}_{nn}$ and the $\text{CLL}_d$ phases. In Fig. 7 (b), we report

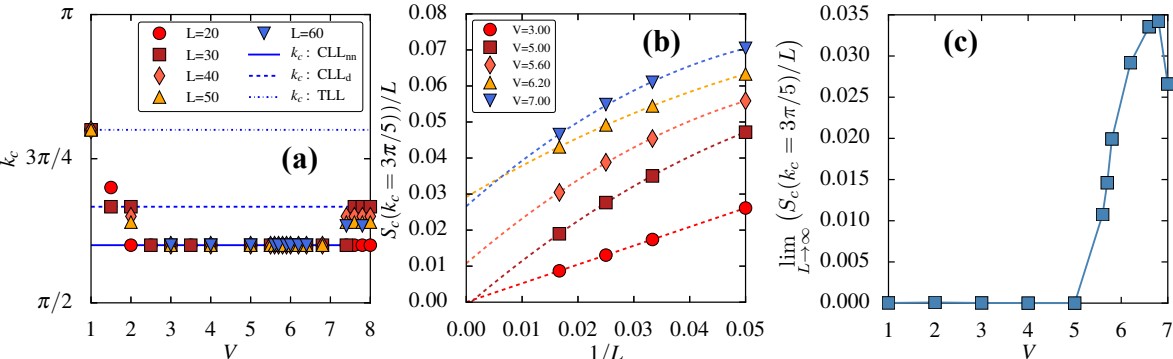

Figure 5: Panel (a) shows the position of the maximum momentum peak of $S_c(k)$ vs $V$. Lines are guides for eyes and correspond to the theoretical prediction of the momentum peak for the TLL (dash-dotted), $\mathrm{CLL_{nn}}$ (full) and $\mathrm{CLL_d}$ (dashed). Panel (b): Finite size scaling of the density-density structure factor $S_c(k_c)$ for different values of $V$. Extrapolation for $L \to \infty$ is obtained by fitting the numerical data with $b + c/L + d/L^2$ (dotted lines). The extrapolated values are shown in panel (c). All simulations are performed at $U = 10$.

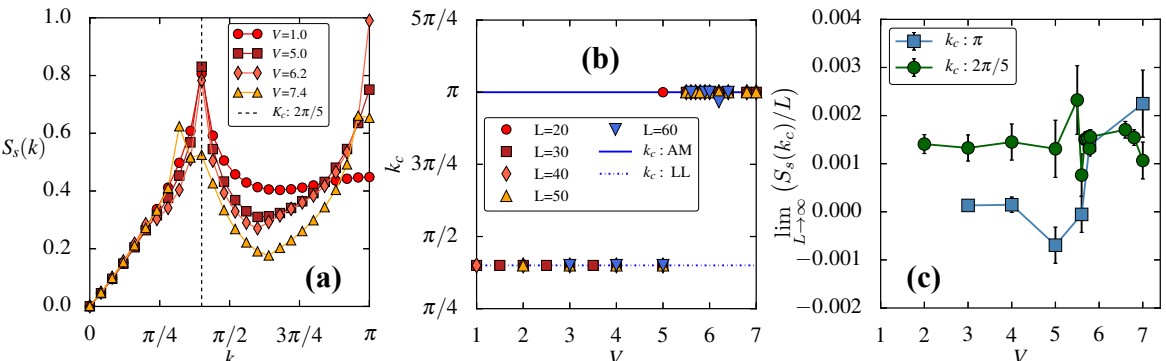

Figure 6: Panel (a): Spin-spin structure factor evaluated at $U = 10$ for different interaction strength $V$. Panel (b) shows the position of the maximum momentum peak of $S_s(k)$ vs $V$. Lines are guides to the eye and correspond to the theoretical prediction of the momentum peak for the LL (dash-dotted), $\mathrm{CLL_{nn}}$ (full) and $\mathrm{CLL_d}$. Panel (c): The extrapolated values of $S_s(k)$ in the infinite-size limit are presented for $k_c = \pi$ and $k_c = 3\pi/5$.

the critical $V_c(L)$ extracted from the maximum of the first derivative $\mathrm{d}(d)/\mathrm{d}V$ as a function of $1/L$. We then extrapolate the $V_c$ in the thermodynamic limit by doing a fit of the form $\frac{a}{L} + b$. We obtain $V_c \approx 6.85$, in agreement with the semi classical prediction $V = 2U/3 \approx 6.66$.

Further insight can be obtained by looking at the ground state entanglement properties of the system. We consider the bipartite von Neumann entropy and extract the central charge of the system according to formula (12). As said before, the numerical calculations are very challenging: due to large frustration, convergence is very slow and size effects are very strong (cf. appendix). Here we summarize our results in Fig. 8, where we show the infinite size extrapolation of the central charge for different values of $V = 1 - 8$. From a detailed analysis of the scaling, we can conclude that:

i) the LL phase has $c = 2$, as it is expected from bosonisation which, for small values of $U$

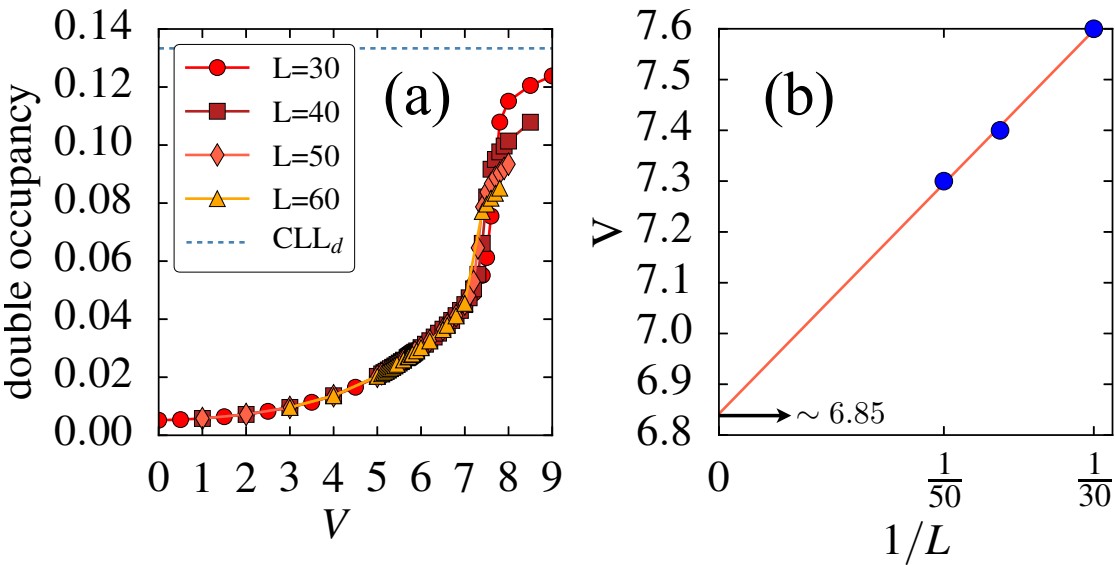

Figure 7: Panel (a) : the double occupancy ($d$) versus $V$ for an on-site interaction $U = 10$. We see that $d$ increases with $V$, a jump occurs at $V \sim 7.5$, indicating a possible first-order transition. The dotted blue line is the expected doubly occupancy within the $CLL_d$ phase. Panel (b): finite size analysis of the critical point extracted from the maximum of the first derivative $d(d)/dV$.

and $V$, predicts a liquid of two bosonic species, with spin and charge separation;
ii) the $CLL_{nn}$ phase has also $c = 2$ (cf. caption in Fig. 8 and appendix): the liquid is made up of clusters with only single occupied sites, with two species of fermions;
iii) the critical point separating the LL and the $CLL_{nn}$ phases is located at $V_c \simeq 5.7$, where the central charge jumps to $c = 5/2$;
iv) even if numerical difficulties limit our simulations to values of $V$ not larger than 9, we see a second critical point $V \simeq 7.4$ at which the central charge is very high.

In order to understand the properties of the different low energy degrees of freedom, we calculate the gaps in the $CLL_{nn}$ phase. Both the charge and the spin gaps are zero in the thermodynamic limit, showing that indeed we are in a massless phase in both the spin and charge sectors. As an example, the finite size behaviour of the spin gap is shown in Fig. 9. On the contrary, we can observe the opening of the single particle gap, as shown in Fig. 10. We use the formula $\Delta \sim (V - V_c)^\nu$ in order to extract the critical exponent at the transition point. In the inset of Fig. 10 we show our numerical data, from which we can extract the estimate $\nu \simeq 1$, a value suggesting a transition belonging to university class of 2D Ising model, which is compatible with the central charge behaviour seen before.

We come now to the interpretation of the numerical results that we have reported above. First of all, let us notice that they are consistent with the fact that spin and charge degrees of freedom are separated at all values of the interaction parameters, so extending the prediction of (standard) bosonisation which describes the TLL phase.
Second, we also see that they are in agreement with what happens for the single species case [37]. In the latter case, both the TLL and the CLL phase are characterised by a central charge

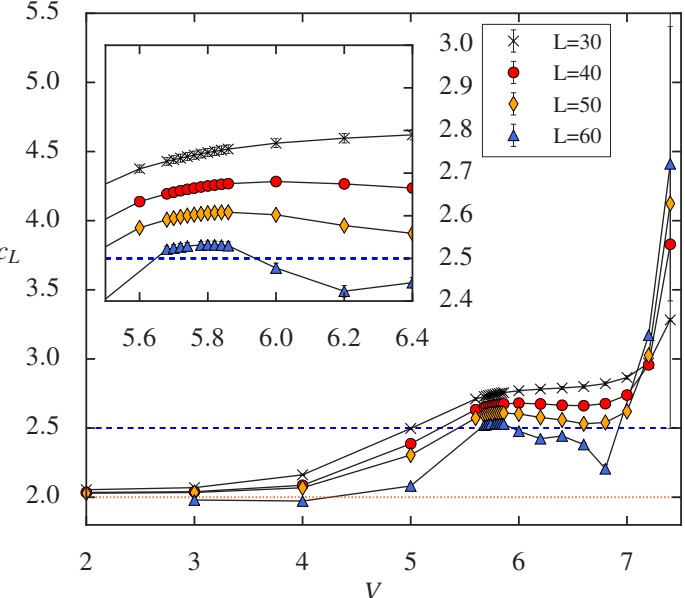

Figure 8: Central charge obtained from formula (12). The data show a phase transition at $V_c \simeq 5.7$ and a possible extended critical region up to $V \sim 7$ (inset). For clarity, in the appendix, we present an in-depth analysis of a point inside the critical extended region, showing that $c_L$ goes back to 2. The critical point is characterised by a central charge that in the thermodynamic limit is given by $c = 5/2$ (the blue dotted line is a guide for the eye). A second transition is predicted for $V \simeq 7.4$.

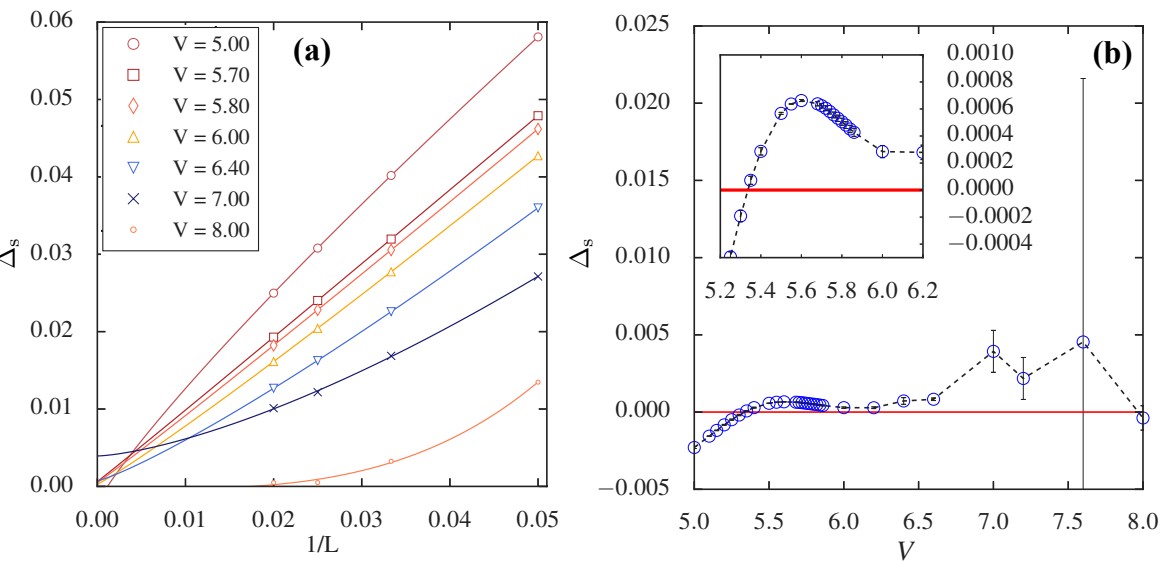

Figure 9: Panel (a): Finite-size scaling of the spin gap for different magnitude $V$. Lines are the best fit of the form $a_1/L^{a_2} + a_0$, with $a_0$, $a_1$ and $a_2$ constant. Panel (b): The thermodynamic extrapolation of the single particle gap as function of the interaction strength. Errors are estimated with the least-square method and are of the order of $10^{-2}$ for large $V$.

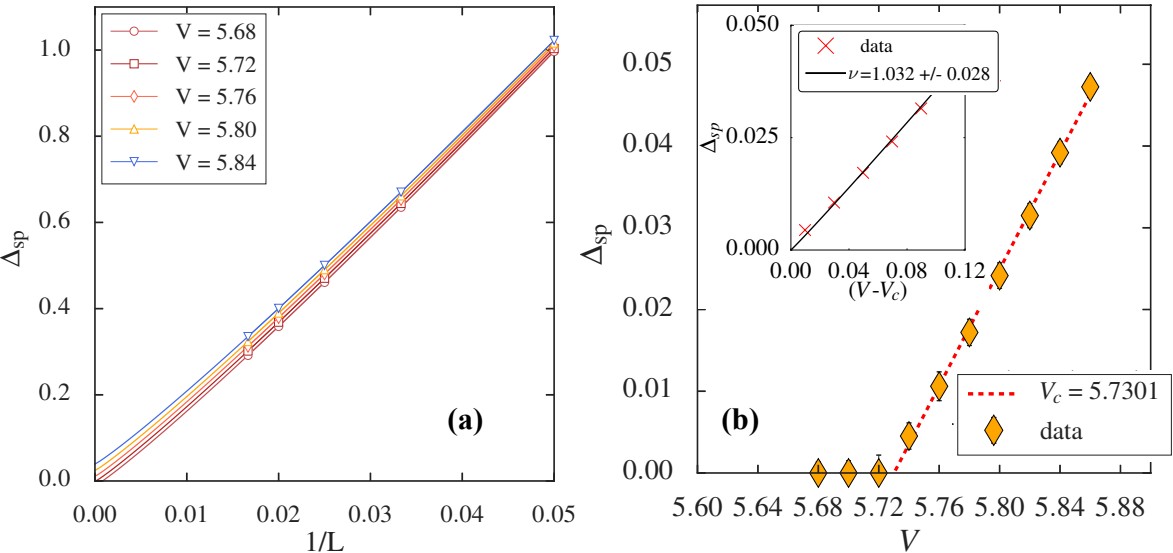

Figure 10: Panel (a): Finite-size scaling of the single particle gap near to the transition point. Lines represent a best fit of the form $a_1/L^{a_2} + a_0$, with $a_0$, $a_1$ and $a_2$ constant. Panel (b): The thermodynamic extrapolation of the single particle gap as function of the interaction strength. The red dotted line is a linear fit in the vicinity of the transition point; its intersection with the horizontal axis yields the critical point $V_c \simeq 5.73$. Errors are estimated with the least-square method and are of the order of the marker size. The inset shows the estimation of the critical exponent $\nu$ at the LL-CLL$_{nn}$ transition point.

$c = 1$. The extended Hubbard model we consider here contains two species of fermions and we find for both CLL phases a value of the central charge $c = 2$. Also, at the TLL-CLL$_{nn}$ transition point ($V_c \simeq 5.7$), the value of the central charge is enhanced by a factor of $1/2$, signalling that an additional (real) fermionic degree of freedom is becoming massless. For the single particle case, this sudden increase of the central charge was interpreted [37] as a signal of an emergent supersymmetry between the compactified bosonic degree of freedom of the liquid phase and of an Ising fermionic one that becomes massless at the critical point. We can confirm that this is what happens also in our case by looking at the sound velocities of the bosonic and fermionic modes at the transition. We can define the sound velocities $v_c, v_s, v_{sp}$ respectively according to [57]:

$$\Delta_c/2 = \frac{2\pi v_c d_c}{L}$$
$$\Delta_s/2 = \frac{2\pi v_s d_s}{L} \tag{15}$$
$$\Delta_{sp} = \frac{2\pi v_{sp} d_{sp}}{L}$$

where $d_\alpha$ is the conformal dimension of the corresponding vertex operator in the conformal field theory that describes the low-energy continuum limit of our model. Notice that we put a factor $1/2$ in the formulae for the charge and spin gaps because their definition implies the action of two vertex operators. The charge and spin bosonic modes are each described by a $c = 1$ conformal field theory, which is completely fixed by the Luttinger parameter $K_{c,s}$ through the formula: $d_{c,s} = 1/4K_{c,s}$ [2,8]. The $SU(2)$ spin symmetry implies that we should assume $K_s = 4$. If we also assume so for the charge sector $K_c = 4$, then we can say that:

$$v_c = \frac{4\Delta_c L}{\pi}$$
$$v_s = \frac{4\Delta_s L}{\pi} \tag{16}$$

For the single particle gap, we use instead the conformal dimension of the first vertex operator in the Ising model, $d_{sp} = 1/8$, to get:

$$v_{sp} = \frac{8\Delta_{sp} L}{\pi} \tag{17}$$

As shown in Fig. 11, our numerical data confirm that all sound speeds become the same at the TLL-CLL transition point, which suggests an emergent superymmetry, i.e a symmetry between bosons and fermions.

As a final remark, we want to comment on the value of the central charge at the second transition $V \simeq 7.4$. Here numerical simulations are particularly hard and we can see from Fig. 8 that we have not reached convergence yet, since $c(L)$ is still increasing very fast with the size of the system. Thus, within the accuracy of our data, we are not able to say whether it is converging to a finite value. A divergent central charge might be the signal that this transition is first order, as it is predicted by semi-classical considerations.

## 3.4 The phases for small values of $U$

In the following, we consider the case of small on-site interaction by fixing U = 1.5. We start our analysis by looking at the charge structure factor $S_c(k)$ for several magnitudes of the

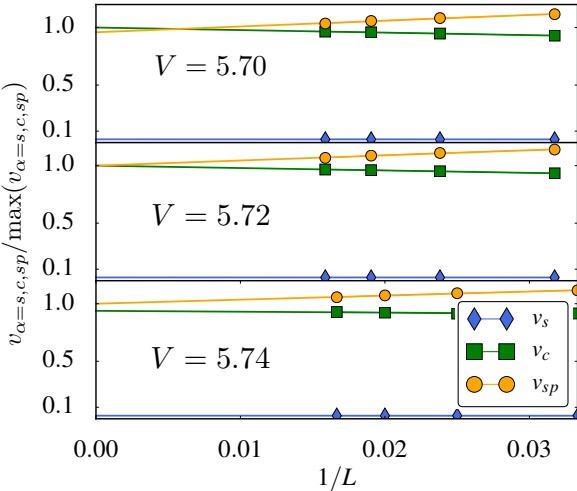

Figure 11: Rescaled sound velocities $v_\alpha^r = v_\alpha/v_m$, where $v_m$ is the maximum of the three sound velocities in the $L \to \infty$ limit, as extracted from the low-energy spectrum.

interaction $V$, presented in Fig. 12 panel (a). We observe (i) the usual TLL phase (no peak) at small $V$, and, (ii) surprisingly, the emergence of a new peak at intermediate value of $V$, see e.g $V = 4, 7$ in Fig. 12.

To understand in-depth the nature of the phase, we perform a finite-size scaling of the peak of the charge structure factor $S_c(k)$. Fig. 12 panel (b) shows examples of the scaling, where solid lines are linear fits. Our data show that $S_c(k_c)$ goes to zero in the thermodynamic limit, signaling a liquid phase. Furthermore, as shown in Fig. 12 panel (c) the double occupancy increases with the interaction strength of the interaction parameter $V$. We interpret these combined results as an emergence of a "new" liquid phase where particles have the tendency of forming pairs, which we call $TLL_d$. Finally, in Fig. 13, we present the central charge, extrapolated in the thermodynamic limit, as a function of the interaction strength $V$. While for large $V$, the frustration prevents any real conclusion, at small and intermediate $V$, the central charge is equal to 2, as expected for a TLL.

We notice that $TLL_d$ and $CLL_d$ appears to be distinct phases. In both cases, due to the small value of the on-site interaction, clusters with doubly occupied sites ($|200\rangle$) are favored with respects to the ones with nearest neighbors. However, qualitatively in the $TLL_d$ region, since $U$ are on the same order of $t$ ($U \gtrsim t$), extra pairs and thus vacant spaces are easily formed. This effect provides higher mobility for the single-particles via first or second-order hopping processes. In contrast, in $CLL_d$, $U \gg t$ and the system tries to minimize every cluster's formation. In this case, the mobility of clusters can only be assured by higher-order hopping processes as in [37].

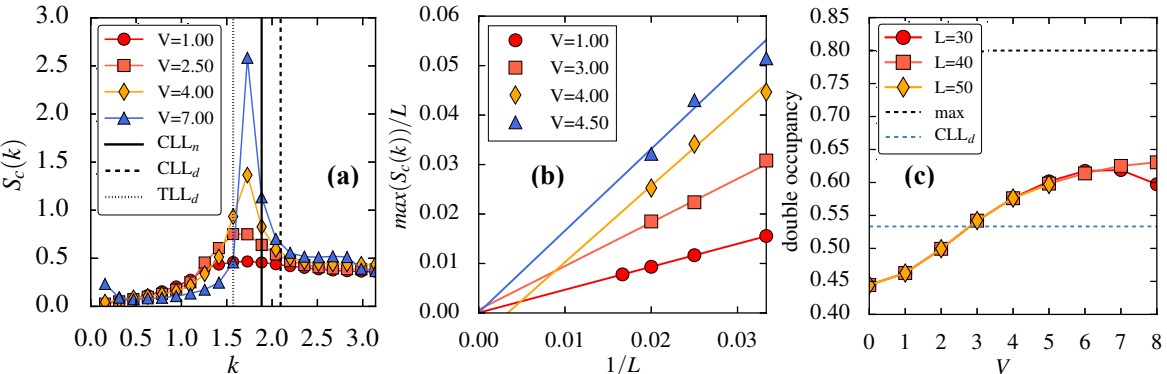

Figure 12: Panel (a): Density-density structure factor for a system size $L = 40$. The black solid line indicates the semi-classical prediction for the $CLL_{nn}$ phase, the dashed one for the $CLL_d$ phase and the dotted line for the $TLL_d$. In panel (b), examples of the finite-size scaling for the peak are shown, where the solid lines are a linear fit of the form $\frac{a}{L} + b$. In panel (c), we present the double occupancy for different values of $V/t$. The dashed lines are guide for the eye and represent the maximum ratio of double occupancy for the fix density $\rho = 2/5$ (black) and the expected value in the $CLL_d$ phase (blue), respectively.

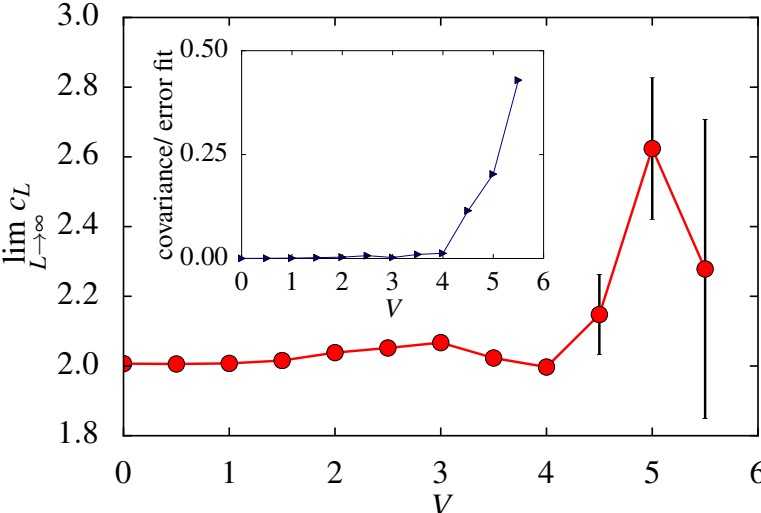

Figure 13: The extrapolated central charge obtained from the Cardy-Calabrese formula Eq. 12 for $U = 1.5$ for different values of $V = 0 - 6$.

# 4 Conclusion

In this work, we have studied in details the ground state phase diagram of a 1D Hubbard model, where particles interact via soft-shoulder potential.

In the first part of the study, we have focused on the classical limit of our model ($t = 0$), for which we have described the frustration mechanisms at play. We have notably shown that these effects lead to the appearance of two cluster type phases, one already known as the

cluster luttinger liquid (CLL) and a second one made of on-site pairs ($CLL_d$). Interestingly, these phases are not captured by the standard TLL theory, and we have found that the transition between the $CLL_{nn}$ and $CLL_d$ occurs at $V = 2U/3$ at this classical level.

In the second part of this work, using the DMRG method, we have studied the phase diagram of our quantum ($t \equiv 1$) model in the whole range of the parameters $U$, $V$ by analysing in detail the properties of the different phases. We have notably demonstrated that the standard TLL phase is the ground state for low values of $U$, $V$. At large $U$ or at large $V$, we have confirmed the existence of the cluster phases predicted classically. Then, focusing on an intermediate value of $U$, we have characterized in more details the different phases and the transition between them. In particular, we have shown that all the phases present in our model are characterized by a central charge $c = 2$, consistent with the separation of the spin and charge degrees of freedom in the TLL theory. However, in the $CLL_{nn}$ we have shown that the single-particle excitation is gapped; hence we have interpreted this phase as a TLL made of composite clusters particles. At low on-site interaction $U$ and intermediate $V$, we have found the presence of a liquid phase characterized by the formation of more one-site pairs as compared to the standard TLL, which we have qualitatively attributed to a strong competition between the tunneling and the on-site repulsion.

At the critical point between the conventional TLL and the $CLL_{nn}$, we have carried out an extensive investigation of the entanglement entropy and the low-lying energy degree of freedom, providing evidence of an enhance of the central charge to $c = 5/2$ compared to the usual TLL, indicating an emergent supersymmetry. Regarding the last point, we have confirmed that the renormalized sound velocities of the emergent bosonic and fermionic modes are indeed equivalent at the critical point within numerical accuracy. Finally, we have shown numerically that the classical prediction for the transition between the $CLL_{nn}$ and $CLL_d$ holds. It would be an interesting question to investigate the quantum phase transition in a two-dimensional model, in a search of exotic quantum phase such as frustration-induced super-stripes [58] superglass [59] and emergent gauge fields [60].

# Acknowledgements

We are grateful to G. Magnifico and D. Vodola for fruitful discussions. G. P. acknowledges support from ANR "ERA-NET QuantERA" - Projet "RouTe" (ANR-18-QUAN-0005-01), Labex NIE and USIAS. E. E. is partially supported through the project "QUANTUM" by IstitutoNazionale di Fisica Nucleare (INFN) and through the project"ALMAIDEA" by University of Bologna. R.C. is supported through the project QUANTUM of INFN.

# A  Appendix: Details about the numerics

In this appendix, we give an insight into the real computational challenge inherent to frustrated systems considered in this work. Indeed, numerical results are hard to extract with high precision, even with the state of art of techniques such as DMRG [53, 55]. In this work, we use a DMRG code provided by ITensor [52], and we impose anti-periodic boundary conditions to reduce boundary effects, keeping up to 9000 states per block and up to 20 sweeps. Furthermore, in order to keep commensurability with the cluster structure ($\mathrm{CLL}_{nn}$), we considered only chains of size $L = 10, 20, 30, 40, 50, 60$. For comparison, in the usual short-range Hubbard model convergence is reached with few hundred states and few sweeps.

We now give an example of the problems we encountered, by showing how we tackled the problem of extracting the central charge. As an instance, we consider the point $U = 10, V = 6.4$, which lies inside the $\mathrm{CLL}_{nn}$ phase. Here, we compare the results of the entanglement entropy by varying some numerical parameters, such as the bond dimensions and the block length of Eq. 12. Using this approach, we are able to extrapolate the central charge in the limit of infinite bond dimension and in the thermodynamic limit. As we will see, the data strongly suggest that the central charge is equal to 2.

In Fig. 14 panel (a), we show the central charge versus the minimum block length $\ell$ used in Eq. (12) for various $D$ and fix $L = 50$. The strong dependence on the bond dimension and the block length is evident. In particular, we find that keeping tiny blocks in the Eq. (12) leads to an overestimation of the central charge, signaling the importance of the non-universal effects. It is possible to estimate the central charge in the limit of infinite number of local states by fitting with a function $c(1 - be^{-aD})$, see e.g. Fig. 14 panel (b). This limit is reported as the black line in Fig. 14 panel (a).

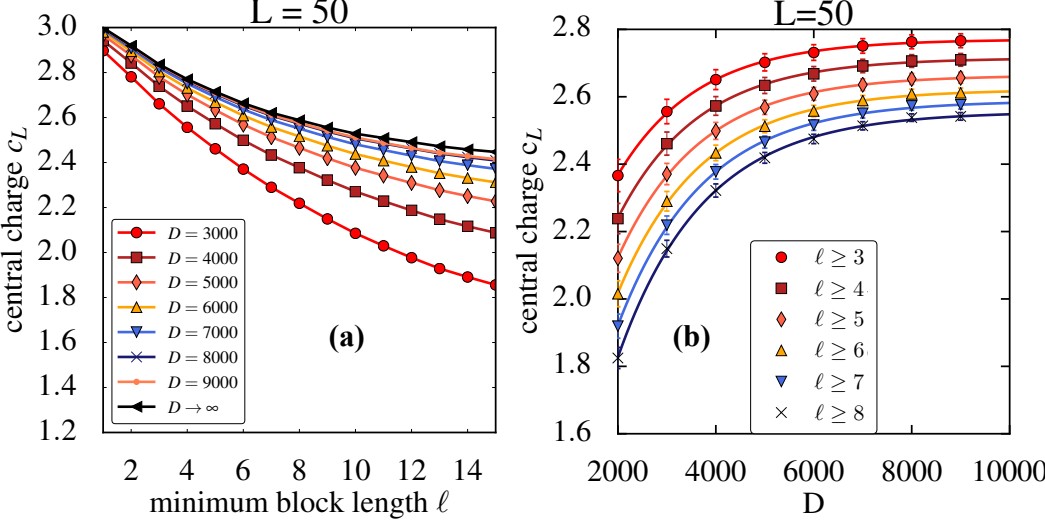

Figure 14: Panel (a): central charge as function of the different block length kept for different local bond dimension. The black line corresponds to an extrapolation in the infinite local states limit. Panel (b): central charge versus $D$, the solid lines are fits of the form $c(1 - be^{-aD})$, providing an extrapolated value of $c_L$ in the limit of infinite $D$.

In order to calculate the central charge in the thermodynamic limit, we now perform a

finite-size analysis at fixed block length $\ell \geq 7$ to avoid non-universal effects. In Fig. 15 panel (a), we plot the central charge as a function of $1/L$ for different bond dimensions. Then, we extrapolate $c_{L \to \infty}$ with a fit of the form $c + \frac{a}{L} + \frac{b}{L^2}$. As mentioned before, the black line corresponds to the infinite bond limit and gives us a first estimation for the central charge around $\sim 1.8$. Numerically, we find here that convergence is particularly hard to reach. The value of the central charge is still underestimated by considering 9000 states per block. Finally, in Fig. 15 panel (b), we show $c_{L \to \infty}$ as a function of the local states $D$. By fitting with a function $c(1 - be^{-ax})$ (blue line), we obtain a second estimation of the central charge in the limit of infinite size and infinite local states: $c \sim 2.3$. These combined results indicate a central charge equal to 2 in the $\mathrm{CLL}_{nn}$ region.

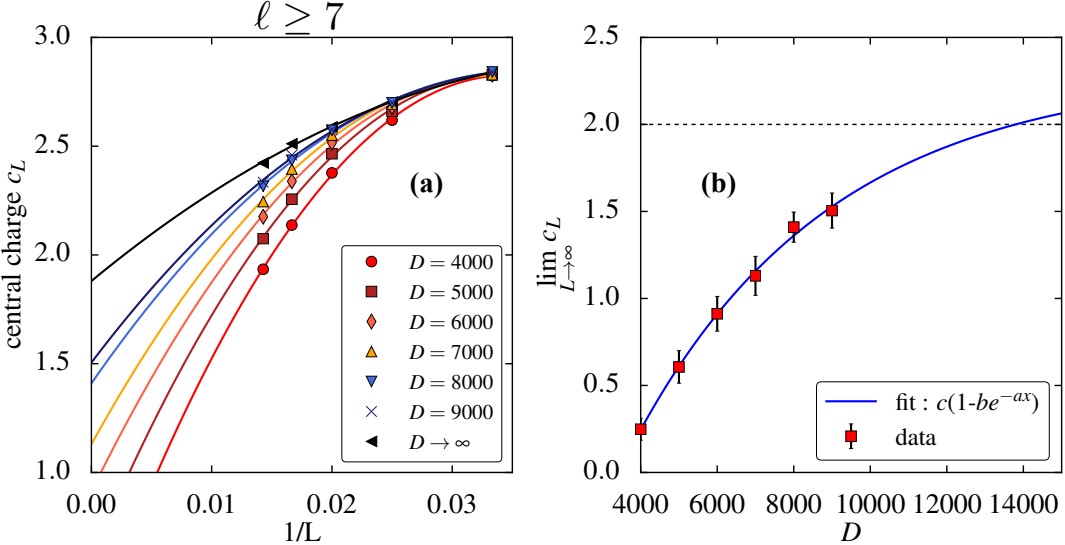

Figure 15: Panel (a) is the finite-size scaling, for block length $\geq 7$, for different bond dimension (colours cf. legend). As previously the black line is the infinite bond dimension limit. In panel (b), we present the extrapolated central charge as function of the local bond dimension.

# B  Appendix: Entanglement entropy

In this appendix, we give some details on how we extract the entanglement entropy and, thus, the central charge. Furthermore, we comment on the convergence of the energy and its variance in different parts of the phase diagram.

In order to extract the central charge, we proceed as follows. We first compute the bipartite entropy for different block lengths and fit via the Cardy-Calabrese formula Eq. 12 to obtain the central charge. Examples are shown on Fig. 16 (a) for different system sizes and for $V = 5.74$, a value close to the critical value $V_c = 5.73$. Let us recall that this formula is valid for $L \gg \ell \gg 1$ [56], otherwise additional finite-size non-universal corrections might arise. Here due to numerical limitation and the commensurability constraint, we have access only to system sizes of $L = 10, 20, 30, 40, 50, 60$. Thus, for sure finite-size effects are strong resulting -as we will see- in an underestimation of the central charge.

In Fig. 16 (b), we show the central charge as function of $1/L$ for different local bond dimen-

sions. The lines are the best fit of the form $a/L^2+b/L+c$, where we considered size corrections up to second-order (similar results are obtained with first order corrections). In the figure we (arbitrarily) choose a minimum block length $\ell = 7$. The figure shows that by increasing the system size $L$ and the local bond dimension to large values $D \gg 1000$ the central charge approaches the value $c = 2.5$, expected from conformal field theory arguments (see main text). Finally, in Fig. 16 (c) the red points show the values of the central charge extrapolated in the thermodynamic limit as a function of the bond dimension $D$. Again, we see that the central charge increases with $D$. The blue line is a fit of the form $c(1 - be^{-aD})$. The extrapolation of $c$ for large bond dimensions shows a good agreement with the value $c = 2.5$.

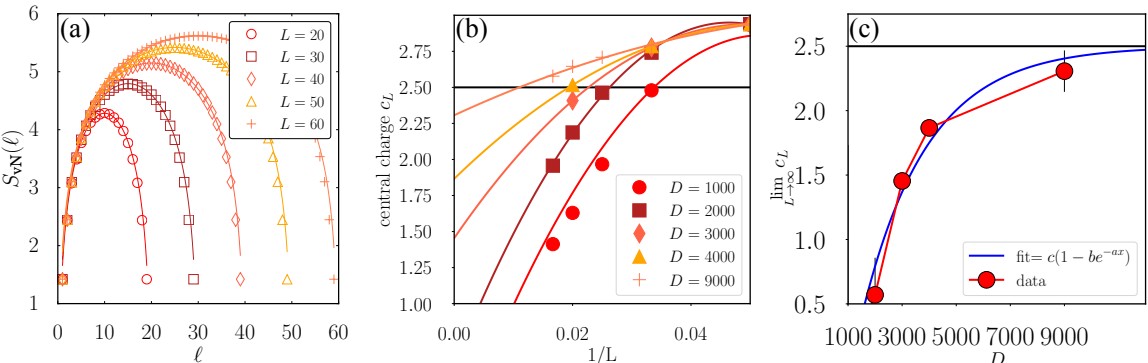

Figure 16: (a) Bipartite entanglement entropy as function of the block length $\ell$ for $V = 5.74$. Data for different sizes are indicated by markers (see color code), while lines are best fit of the form of Eq. 12, where $c$ the central charge and $a_0$ are optimised parameters obtained via a least-square methods. (b) Finite-size values of the central-charge obtained by fitting Eq. 12 for minimum block length $\ell = 7$. Lines are best fit of the form $a/L^2 + b/L + c$. (c) Extrapolated central charge as function of the local bond dimension, the blue line is a fit of the form $c(1 - be^{-aD})$.

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
