# Peer review of "One-dimensional extended Hubbard model with soft-core potential"

_SciPost Physics_

## Round 1 · Referee Report · Anonymous · 2019-10-25

Strengths

1- Numerical study of an extended Hubbard model in one dimension, relevant to some cold atoms experiments.
2- Rich ground-state phase diagram including Tomonaga-Luttinger liquid (TLL) and cluster Luttinger liquid (CLL)

Weaknesses

1- Most numerical raw data are not presented.
2- The phase diagram is only sketched.
3- The cluster Luttinger liquid (CLL) is not a new phase and has been proposed and found long time ago. References should be included. The additional fermionic mode at the transition has already been presented by some of the authors in a different model. As such the paper lacks novelty to some extent.

Report

This paper provides mostly a numerical investigation of the ground-state phase diagram of some extended Hubbard model in one dimension. Using standard measurements and DMRG for numerical simulation, a sketch of phase diagram is presented for a fixed density and potential range. In principle, we could expect to have a large part of the phase diagram studied ?

Regarding numerical simulations, the authors have chosen to use antiperiodic boundary conditions, which is known to be hard for DMRG. What is the reason for such a choice ? Apparently this strongly limits the sizes that can be simulated: typically, data are presented only for L=30 or L=50, which is quite small to discuss criticality and provide accurate results. Even though an Appendix does provide some details about simulations, I would like to ask the authors to provide more data, in particular about the entanglement entropy. Indeed the extraction of the central charge seems quite involved and inaccurate, which puts some doubts on the claim of having c=5/2 at the transition, see Fig. 6. In the same figure, it looks extremely strange to have an effective c diverging. I expect that for an SU(2) system with a single charge channel, the maximum of c would be 3+1.

Moreover, I am not very fond of the way that the CLL is presented. In the abstract, it is written that it was first predicted in Ref. [1]. But in the abstract of Ref. [1], it is explicitly written that it was "first predicted" in Ref. [37]... To my knowledge, it is in fact a much older concept, found for instance in multicomponent fermionic chains (Lecheminant et al. PRL 95, 240404 (2005); Roux et al EPJB 68, 293 (2009) etc.) . In fact, even fermionic pairing on a correlated t-J ladder can be seen as a mechanism to go from a Luttinger liquid made of charge-1 objects to a Luther-Emery phase made of charge-2 pairs.

Minor point: the definition of the Luttinger parameter K does not seem to be the standard one (K=4 for SU(2) symmetry) ?

In conclusion, I do not think that the paper can be published in SciPost given its weaknesses, both in terms of physical findings (which are merely incremental) and in terms of data presentation and analysis.

---

## Round 1 · Referee Report · Anonymous · 2019-10-31

Report

In the present manuscript, titled “One-dimensional extended Hubbard model with soft-core potential”, authors Thomas Botzung, Guido Pupillo, Pascal Simon, Roberta Citro and Elisa Ercolessi study the ground state properties of a particular Hubbard model with specific long-range interactions termed “soft-shoulder potential”, which extends over several sites without decay, in addition to local repulsion between spin-up and spin-down electrons. The authors present results arguing for the emergence of Cluster Luttinger Liquid (CLL) phases in this model at a particular density and especially for the emergence of a fermionic excitation branch in addition to the usual bosonic ones at the transition from Tomonaga Luttinger liquid to CLL, which they term “supersymmetric”.

In light of substantial recent experimental activity on quantum many-body systems with long-range interactions in neutral dipolar gases, Rydberg atoms, trapped ions and in particular non-decaying long-range interactions mediated via light fields in optical cavities, the work contained in this manuscript deals with an interesting and timely topic. Thus, the authors findings, if substantiated, would certainly be suitable for publication in SciPost and be of interest to the community. However, in its present form the manuscript requires analytical improvements to substantiate key findings before being suitable for publication. These are listed and discussed in the following, along with several minor points on presentation and legibility:

(1) The most interesting single finding in the present manuscript is the authors claim of the emergence of fermionic excitations along with the standard bosonic ones at the transition between TLL and CLL_nn around V_c = 5.73. The findings the authors provide for this however require a more careful analysis than the one currently presented. On pg. 10 the authors state that they supposedly show the infinite size extrapolations of the central charge together with the finite-size data in Fig. 6, but such extrapolations are actually missing from the figure.
Should the authors supply the extrapolations, care should be taken to perform these in a theoretically rigorous manner, based on known analytical results for the system-size dependence of the central charge (there already are several infinite-size extrapolations in the present manuscript that appear to be ad-hoc and not based on known scaling behaviour - see below). This is especially important because the inset in the current Fig. 6 does not necessarily force the reader to the conclusion that c=2.5 is the correct endpoint of convergence. In fact, the size dependence is pronounced when looking at e.g. the final drop from L=50 to L=60, which leads to significant doubts that c=2.5 really is the infinite-size value.
On the same topic, on pg. 12 the authors state that all sound speeds coincide around the transition, but Fig. 9 directly contradicts that statement, with v_s always far below the other two velocities. Further up, above eq. (8), the authors, just assume that K_c = 4 at the transition without any proof, reference or theoretical argument.

Finally, explaining the nature of the supposed supersymmetric point requires improvement, assuming the claim of c=2.5 when 1/L=0 can be substantiated. How is the claim of supersymmetry (every fermion has a boson partner) justified when there are two bosonic modes but only one (Majorana) fermionic one? MPS-based methods, including ITensor, provide a number of ways to manipulate and perform measurements on wave functions - how could these tools be deployed to directly probe the presumed fermionic excitations at the presumed transition point?

(2) The appendix of the manuscript raises a number of critical technical questions requiring further explanations, on which the methodological soundness of this work hinges. Discarded weights or energy variances are not provided for any calculation. But fitting in either one of these quantities is the actually well-defined procedure in MPS-based methods, and NOT in the bond-dimension D. Fitting in D is a pure heuristic without a theoretical basis, that will usually give wrong results when compared against energy-variance or discarded weight extrapolations.
Also, many results are obtained for large U- and V-values (up to 50t), which can lead to very slow convergence requiring many sweeps to allow the particles sufficient opportunity to tunnel past each other - no checks are provided that 20 sweeps are actually enough to accomplish this here.
Then there is the issue of the block lengths used in the fit to eq. (6) (if fitting is actually what is being performed in e.g. Fig. 12 - this is also not clearly explained). On the left of Fig. 12, block lengths up to 15 appear to be included, but on the right only up to 8. And why is the left-hand side showing six different fitting-ranges (upper limits from 3 to 8), instead of just the biggest fitting range, l<=8? What are these “non-universal effects” vaguely invoked for both the supposedly “tiny blocks” (bottom of page 16), but also for blocks with l>=8 (top of page 17)? If these effects are present at l=8, why are they included on the left-hand side of Fig. 12?

(3) In a number of places in the manuscript, purely heuristic fitting procedures are used, seemingly or actually without theoretical support. This is most noticeable in the fits for the spin gap and the single particle gap, Figs. 7 and 8 (left), where a generic power law with three fitting parameters is used to fit 4 or 5 points. As the errors of the initial spin-gaps are seemingly not included in the error of the extrapolation, the chosen fit is almost always delivering a rather small least-square error - unsurprisingly.
Fig. 4 panel (b): 4 points fitted by a+b/L+c/L^2 (from the main text, not mentioned in the caption). If this is motivated by theory, it is not explained.
Fig. 8 panel (b): Is the red dashed line a linear fit of all data points? If not, why?

(4) In the abstract the authors write „[…] similarly to what first predicted in a hard-core bosonic chain in Ref.1“, but in Ref. 1 we found: „[…], we demonstrate that the CLL phase [first predicted in Ref. 37] is separated from a conventional TLL [...]“. Why are the authors referring to Ref. 1 instead of Ref. 37?

(5) The usual (minimalistic) references for DMRG are White 92, 93 and Schollwöck 05 and for MPS (of which ITensor is making extensive use) Schollwöck 11.

(6) Soft-core [six occurences] vs. soft-shoulder [5 occurences]: the authors seem to use these terms interchangeable. This might be confusing, sticking to either one of these formulations may improve legibility.

(7) The authors might consider to add a sketch of the soft-shoulder potential as was provided in Ref. 1 and Ref. 37.

(8) Figure 1 is poorly explained in the current state. As the supposed overview figure of the manuscript, it should provide a comprehensive overview to the reader. It does not do so when the phase abbreviations are not even explained in the caption text, and the position of the supposed supersymmetric transition point is not marked and explained as such - is the red dot, which currently has no explanation at all, meant to denote the transition?

(9) On the bottom of page 4: „Finally, in the appendix, we give some details on the numerics and draw our conclusions“; As the conclusions are given before the appendix, this sentence should be corrected.

(10) Below eq. (1) the various number operators are not defined. And as a side note: index notation is inconsistent, sometimes indicies are separated by a comma and sometimes not.

(11) Check label and axis sizes. E.g., the inset of Fig. 8 panel (b) is extremely small.

(12) Below eq. (3), the definitions of spin-up and spin down numbers are incongruous with the rest of the manuscript: probably the first equals sign should be a plus, leading to N=N_up+N_down?

(13) Right below eq. (4), the definition of the spin-imbalanced ground state energy is potentially confusing. To ensure consistency with eq. (4), it would need to add $\pm 1$ to the subscript.

(14) Typo in 2.1 2) „[…], thus a phase transition form standard TLL to a new [...]“; form -> from.

(15) Last paragraph on page 7: „[…] in the appendix 4.“ should be appendix A.

(16) In 3.2 in the description of Fig. 4 panel (b): The lines are dashed not dotted.

(17) In 3.2 the authors introduce yet another single-particle density n, after having introduced N, N_up, N_down, and <n_up>, <n_down>, and rho. This may lead to confusion on the part of the reader.

(18) In 3.3 „Furthermore, as shown in Fig. 10 panel (c) the double occupancy increases with the interaction strength.“ It should be clarified which interaction parameter exactly is meant here - currently only the figure on the following page allows the reader to deduce that V is meant.

(19) Caption of Fig. 11: „[…] obtained from the Cardy-Calabrese formula […]“. Add a reference to either Eq. (6) or Ref. 53 or both.

(20) ITensor is spelled improperly in the appendix.

(21) In figures throughout the manuscript Hamiltonian parameters are sometimes scaled with t and sometimes not when used as axis labels. Furthermore, some axis labels are missing altogether, e.g., Fig. 7.

---

## Round 1 · Referee Report · Natalia Chepiga · 2019-11-1

Report

In the present manuscript the authors study an extended Hubbard model with nearest- and next-nearest-neighbour repulsion with equal couplings. The authors reports the appearance of the so-called cluster Luttinger liquid phases and discuss the transition between them.
The problem itself sounds very interesting and relevant, and after substantial revision the manuscript can be suitable for publication in SciPost. However at this stage the presentation of the numerical results is very poor and sometimes misleading. In particular, the very existence of the critical line between two types of CLL has not been justified. Can it be a crossover with varying size of the supercell (equivalently with varying k) as a function of coupling U and V? The possibility of the first-order transition suggested by the semi-classics has not been excluded as well. Analysis of the numerical data is not complete and in not reproducible, due to missing details (see comments below). Therefore I do not recommend the manuscript for publication in its current form.

My detailed comments and questions are listed below:

C0: I would be curious to see the plot of the wave-vector k as a function of U and V coupling constants. I am asking for something similar to Fig.4(a) but, perhaps with better resolution around V=1.5 and V>7; Since both CLL phases are critical one can easily imagine that at an intermediate value of the coupling constants, a competition between the two leads to a new state with an intermediate values of the wave-vector k. Of course, accurate analysis would require larger system sizes. But there is already a very good indication in favour of incommensurate values of q - in Fig.5(a) peaks for V=7.4 are off their commensurate values. Also, incommensurate peaks have been reported in Fig.10(a). However, finite-size scaling in panel (b) has been performed only for V<=4.5 and only with three data-points, which might not be sufficient in the presence of incommensurability. In this respect it might be instructive to do the calculation of the chain with L which is not multiple of 10.
C1. Fig.1: V=2U/3 does not go to zero; red line corresponds to semi-classical results, and it is not actual phase boundary, perhaps it should be mentioned in the caption, or just replaced by the actual boundary. Red dot is not mentioned
C2. Eq.(1) sum over \sigma in the first and last term is missing
C3. Eq.(3) 2nd line, +/- in the fist term is missing
C4. Eq.(6) is valid for periodic boundary conditions, while in the appendix we find:
"... and we impose anti-periodic boundary conditions to reduce boundary effects". In this respect I have several questions:
Q1: Which kind of "boundary effects" are avoided by choosing anti-periodic over periodic boundary conditions;
Q2: Is Calabrese-Cardy formula actually valid for the anti-periodic boundary conditions. It is not trivial at all, so either comment or an appropriate reference is necessary here.
Q3: Fig.6: Eq.(6) apart from the central charge contains a non-unversal constant a_0. How exactly the central charge is extracted. If it has been done in a usual way, by fitting the finite-size profile of the entanglement entropy, an example of the fit should be provided.
Q4: How exactly the anti-periodic boundary conditions have been imposed?
Q5: How the anti-periodic boundary conditions affect the value of the dominant wave-vector k in Fig.3,4,5...
Q6: Filling 1/5 is chosen to stabilize the CLL phase. Is there any experimentally relevant motivation for this value?
C5: "...Here we summarize our results in Fig. 6, where we show the infinite size
extrapolation of the central charge". There is no finite-size extrapolation in Fig.6, but it is necessary. In particular, around V=5.7 the central charge decay so fast, so it is hard to believe it will stay at c=5/2 at the thermodynamic limit. Below Fig.6, "(iii) the critical point... is located at Vc... where the central charge jumps to c=5/2". The "jump" is only due to lack of the point on the way to V=5.7.
C6: I find it quite misleading to provide the unphysical results such as c=4 at V=7.4 in Fig.6; or negative gap in Fig.7.
C7: In Fig.6, results for N=60 and 6<V<7 is very different from those of N<60. Is it due to convergence, or there a deep reason for this, e.g. finite-size crossover in the ground-state.
C8: Fig.7 three-parameters fit with only four data points is not very reliable. Behaviour for V=8 is very different from V<7. It would be interesting to see the value of the wave-vector k at each point. If I am not mistaken, it changes with L.
C9: Eq.7: the operator content depends on the boundary conditions, and therefore d_\alpha for periodic boundary condition is not necessary the came as d_\alpha for an anti-periodic one. Comments here would be relevant.
C10: Fig.9, no y-labels
Q7: In appendix: " In particular, we find that keeping tiny blocks in the Eq. (6) leads to
an overestimation of the central charge, signalling the importance of the non-universal effects." First, it has been mentioned in the original paper Ref.[53], that the Eq.6 is only valid for l,L-l>>1. Second, in Fig.13 l<=7 is used in contradiction with the comment above.
Q8: I wonder, whether the authors consider an option of open boundary conditions. Of course, the price to pay is an emergence of the edge effects, but fixing the boundary conditions one can actually profit from the edges and from the boundary conformal field theory. At the same time, in (anti-)periodic system entanglement is associated with two cuts, and therefore is superimposed, that leads to extremely large bond dimension D necessary to reach the convergence. In open boundary conditions there is only one channel of entanglement, so the bond-dimension grows much slower, and one can compute the wave-function for much larger systems sizes.

---

## Round 2 · Referee Report · Anonymous · 2020-8-3

Report

With their revised version, the authors have improved their manuscript and addressed some of the previous concerns. However, some issues remain open and new ones have been introduced by the revision.

The previous Report 1 pointed out some older references for Cluster Luttinger Liquids (CLLs, see Lecheminant et al., PRL 95, 240402 (2005); Roux et al., EPJB 68, 293 (2009); etc.). One might argue that this is not exactly the same model and thus not of direct relevance. However, the authors write in their reply:

| We have changed the reference in the abstract and added the suggested references in the introduction.

Neither of the two is actually true. If this is just an oversight, this should be fixed.

The second point that was raised previously concerns the issue of boundary conditions. The authors write at the beginning of section 2 and in Appendix A that they have used "antiperiodic" boundary conditions. In their reply, they write

| Open boundary conditions introduce boundary effects that can mask the cluster structure of the ground state in some regions of the phase diagram.

While it is true that open boundaries give rise to additional finite-size effects, they usually require much smaller numerical effort in DMRG, i.e., a much smaller value of $D$ is able to provide quantitatively accurate results than needed for (anti-)periodic ones. So, there is a balance to strike between numerical accuracy and additional finite-size effects. The authors write at the end of their reply to Report 3:

| Indeed, we did try with open boundary conditions, finding that edge effects arising from finite lattices kill completely the cluster formation.

This is a statement that in my opinion would merit integration into the manuscript. One might also wonder why antiperiodic boundary conditions are better than periodic ones. Is this related to the specific choice of filling $\rho=2/5$?

I have some further minor comments:
1- The first paragraph of the Introduction talks about field theories, but then the authors study a lattice model. While field theories do emerge as low-energy long-distance descriptions of lattice models, I think that an explicit statement on the relation would be useful for the reader.
2- The order of the references [52-55] in the Introduction and at the beginning of chapter 3 is strange: they start with a recent library [52] and finish with the original paper [55]. I recommend to restore chronological order (or separate method and implementation).
3- Line 3 of page 4 and line below Eq. (17): "sound speed" -> "speed of sound" (or "sound velocity").
4- What is "soft" about the hard cutoff of the repulsion at $r_c$? I think that the authors should explain the terminology "soft-shoulder" (or change it).
5- Typesetting "$r_c$" versus "r$_c$" should be unified.
6- Typesetting "CLL$_{\rm nn}$" and "CLL$_{\rm d}$" versus "CLL$_{nn}$" and "CLL$_{d}$" should be unified.
7- Page 9, section "Von Neumann entropy": it is strange to have no reference about these concepts until the line before Eq. (12).
8- On page 15, the authors talk about "a factor of 1/2", but I think that they mean an additive rather than a multiplicative constant.
9- In Fig. 11, the blue lines are difficult to distinguish from the black lines separating the different panels. In addition, the statement "our numerical data confirm that ALL sound speeds become the same" on the line below Eq. (17) is obviously not true for $v_s$.
10- The relation of the new Appendix B to the old Appendix A is not clear since they discuss very similar issues. I suspect that the Appendix A concerns the CLL$_{nn}$ phase while the Appendix B concerns the transition from the CLL$_{nn}$ to the CLL$_d$ phase. If this is correct, explicit statements to this effect would be helpful.
11- Even if this can probably be fixed during the production stage, I nevertheless suggest to fix the spelling of the chemical formula "SrCuO$_2$" in Refs. [11] and [14].
12- Ref. [17] duplicates Ref. [1].
13- Even if this can probably be fixed during the production stage, I nevertheless suggest to fix the name "Hall" in Refs. [21] and [22].
14- There are some further minor typographic errors (such as "L" versus "$L$" on line 6 of of the section "Von Neumann entropy", "CLuster" on line 6 of the caption of Fig. 3, "double occupied" versus "doubly occupied" [several instances], "U" versus "$U$" on the first line of section 3.4, etc.) that I hope can be fixed during the production process.

I recommend publication of the manuscript once the above issues have been addressed.

---

## Round 2 · Referee Report · Natalia Chepiga · 2020-8-9

Report

The manuscript has been improved during the revision. In particular, it contains some examples of finite-size and finite-bond scaling that makes the results reproducible and that allows the reader to make own conclusion based on the raw data.

1. My main concern in this new revised version is possible formation of the floating phase – Luttinger liquid phase with non-frozen (varying) wave-vector q. As I already pointed in my first review, there are numerous indication for it: (i) the structure factor has a peak at the intermediate value of k in Fig.12(a) and (ii) with increasing L the peak goes away from its commensurate value 2\pi/3.
The distance between two commensurate values 2\pi/3 and 3\pi/5 is only 0.0667\pi is the same order of magnitude as an error in numerically extracted value of q: \delta q\approx 2\pi/L\approx 2\pi/60=0.033\pi. This definition of error corresponds to an assumption “that L always contains an integer
number of clusters” mentioned on p.11 and the fact that correlation length always diverges.
These simple calculations implies that in order to clearly distinguished between the two scenarios - either direct transition between CLL_nn and CLL_d considered in the paper or the floating phase that I would expect – one has to go to the system sizes an order of magnitude larger than those presented in the manuscript. Let me be more explicit, to be able to measure, say, 10 points between two commensurate values 2\pi/3 and 3\pi/5 one has to do simulations on L>600 sites. Before these simulations are performed I do not see how one can choose one option over the other. So either this puzzle has to be resolved (e.g. numerically) or all possibilities have to be listed.

2. Critical system with (anti-)periodic boundary conditions, L>600 and c>~2 is quite unrealistic task. By contrast, in open systems where the bond dimension D is roughly equal to a square-root of the corresponding bond dimension for periodic one, such system sizes often can be accessed. But here comes the second puzzle – the authors write:
“Indeed, we did try with open boundary conditions, finding that edge effects arising from finite lattices kill completely the cluster formation.”
First, I 100% agree with the first Referee who suggests that this statement should appear in the main text.
Second, if cluster formation is a bulk process, it should appear with open boundary conditions as well. Of course, much larger system sizes or fixed (conformally invariant) boundary conditions might be necessary to overcome the edge effects. But this issue of “complete killing of the cluster formation” definitely requires systematic investigation, to exclude the possibility that CLL_d phase is a finite-size effect that will be ruined in the thermodynamic limit.

I have further minor comments:

p.13 “Finally at strong V , k c is equal to π/3, which corresponds to the CLL_d phase” should be 2π /3

p.12 “the discontinuity in the energy between the CLL_nn”. The energy has to be continuous, do the authors mean discontinuity in its derivative?

Fig. 8. What is the meaning of “a possible extended critical region up to V ∼ 7”. The entire parameter space is critical isn’t it?

p.23 “Thus, for sure finite-size effects are strong resulting -as we will see- in an underestimation of the central charge.” Finite-size effects usually result in over-estimation of the central charge, this can be see n in panel 16(b), c decreases with 1/L. By contrast finite DMRG bond dimension indeed results in underestimated central charge.

In Fig.3 it is not clear how and where the TLL phase is separated from the TLLd phase.

Let me repeat what I already wrote in my first review. I think the problem itself is interesting and relevant. However, there are still important questions that remains open or have been overlooked in the current manuscript. Therefore I cannot recommend the publication of the manuscript before points 1 and 2 will be addressed.

---

## Round 2 · Author Response

Dear Editor,
We apologize for the delay in the resubmission, and thank you and the referees for your patience. We also thank the referees for taking the time to read the manuscript carefully and for providing constructive criticism to improve it.

We appreciate Referee B for her/his positive remark: “ ... the work contained in this manuscript deals with an interesting and timely topic. Thus, the author’s findings, if substantiated, would certainly be suitable for publication in SciPost and be of interest to the community.” Unfortunately, Referee B does not recommend publication asking for analytical improvements and a better presentation. Referee A does not recommend publication due to “ Analysis of the numerical data is not complete and is not reproducible, due to missing details...”. Referee C also points out the lack of structure and the data presentation of the paper.

We understand the points raised by the referees. Thus we have restructured our paper to improve clarity (see list of changes) and answer all raised questions, as detailed in the different detailed answers. The presentation of our results may have been unclear to some readers. Therefore, we adapted our manuscript in several places following recommendations of referees (as detailed below). We now emphasize the classical limits and extent the numerical aspect by notably adding a new appendix section. We are convinced that our modified manuscript is suitable for publication in SciPost journal.

Sincerely,
Thomas Botzung, Guido Pupillo, Pascal Simon, Roberta Citro and Elisa Ercolessi

---

## Round 2 · List of Changes

-We have added a completely new section, “Classical analysis” that considers the exact solution of the ground state in the classical limit. In this section, we show the different possible phases present in our model and demonstrate the crossover between CLLnn and CLLd.
-We have extended the section “observables” by adding more information about the structure factor, entropy, and low-energy degrees of freedom. In particular, we explicitly derive results of low-energy degrees of freedom in the classical limit.
-We have added a new Appendix, where we explain in detail how we extract the central charge from the Cardy-Calabrese formula.
-We have included a new figure in the main text and paragraph to confirm the crossover between CLLnn and CLLd in Sec 3.3. (see Eq. 14 and Fig.7)
-We have modified the outline and the conclusion to include the new section.

---

## Editorial Decision

editor-in-charge_assigned